# Bayesian analyses indicate bivalves did not drive the downfall of brachiopods following the Permian-Triassic mass extinction

Zhen Guo [1], Joseph T. Flannery-Sutherland [2], Michael J. Benton [2] ✉ & Zhong-Qiang Chen [1] ✉

Certain times of major biotic replacement have often been interpreted as broadly competitive, mediated by innovation in the succeeding clades. A classic example was the switch from brachiopods to bivalves as major seabed organisms following the Permian-Triassic mass extinction (PTME), ~252 million years ago. This was attributed to competitive exclusion of brachiopods by the better adapted bivalves or simply to the fact that brachiopods had been hit especially hard by the PTME. The brachiopod-bivalve switch is emblematic of the global turnover of marine faunas from Palaeozoic-type to Modern-type triggered by the PTME. Here, using Bayesian analyses, we find that unexpectedly the two clades displayed similar large-scale trends of diversification before the Jurassic. Insight from a multivariate birth-death model shows that the extinction of major brachiopod clades during the PTME set the stage for the brachiopod-bivalve switch, with differential responses to high ocean temperatures post-extinction further facilitating their displacement by bivalves. Our study strengthens evidence that brachiopods and bivalves were not competitors over macroevolutionary time scales, with extinction events and environmental stresses shaping their divergent fates.

The Phanerozoic evolution of animal diversity has been shaped by the balance of extinction and origination processes underpinned by biotic factors and abiotic drivers, the Red Queen[1] versus Court Jester[2] models, respectively. The interplay of abiotic and biotic factors over different temporal and spatial scales caused the waxing and waning of clades and their replacement by one another[3,4]. Of these, the decline of brachiopods, coupled with the rise of bivalves in the Phanerozoic is a textbook example of clade replacement in palaeontology[5] (Fig. 1). Both clades originated in the Cambrian, survived the 'Big Five' mass extinctions[6,7], and thrive in today's oceans. They have lived in the ocean for >500 million years (Myr) and are two of the most diversified invertebrate clades in marine ecosystems[8,9]. Nevertheless, brachiopod diversity declined dramatically at the point of the Permian-Triassic mass extinction (PTME) ~252 Myr ago, whereas bivalve diversity increased, demonstrating the brachiopod-bivalve switch in their relative richness[5] (Fig. 1).

The PTME coinciding with the brachiopod-bivalve switch also marks one of the largest events in the history of marine life, the switch from Palaeozoic- to Modern-type evolutionary marine faunas[10,11], the most dramatic turning point in the 540 Myr of the Phanerozoic[12]. The rise of the modern marine fauna, dominated by bivalves, gastropods, crustaceans, echinoids, and neopterygian fishes, was also a point of substantial increase in energy capture by marine life mediated by new photosynthesising plankton groups from the Late Triassic onwards. The Mesozoic marine revolution[13] began in the Late Triassic and Jurassic, represented by enhanced arms races between predators and prey and, in turn, the dominant new groups were faster and meatier than their Palaeozoic precursors[14–16].

[1]State Key Laboratory of Biogeology and Environmental Geology, China University of Geosciences (Wuhan), Wuhan 430074, China. [2]School of Earth Sciences, University of Bristol, Bristol BS8 1RJ, UK. ✉e-mail: mike.benton@bristol.ac.uk; zhong.qiang@cug.edu.cn

Owing to similarities in body plan (i.e., two shells) and overlaps in ecology (i.e., similar feeding behaviours, modes of life, and living habitats)[17], brachiopods and bivalves have long been regarded as competitors[17–20]. Gould and Calloway[5] first challenged this view and considered that both groups possessed comparable diversity patterns in the Palaeozoic, and that the PTME reset their initial diversities and altered their relative dominance. In arguing against active competition between members of both clades, they were described as 'ships that pass in the night'[5]. However, debate on the competition between the two clades and their ecological roles in ecosystems continues[18–29]. Moreover, the decline of brachiopods after the PTME has also been attributed to intense predation[30,31], decreased ability to expand habitat distribution[17,32,33], or increased energy flux[34]. Nevertheless, data supporting the previous scenarios mostly relied on the relative richnesses of brachiopods and bivalves, without considering their underlying diversification dynamics (i.e., origination and extinction rates), nor have their relationships with biotic (e.g., self- and predator diversity dependence) and abiotic (e.g., temperature, seawater chemistry, tectonic regulation) factors received substantial statistical scrutiny (but see Liow et al.[20] and Reitan and Liow[27]).

Here we quantify the diversification dynamics of brachiopods and bivalves, along with their drivers, in a Bayesian framework (PyRate[35–37]) to investigate the timing and triggers of the brachiopod-bivalve switch. First, we estimate the origination and extinction rates, and diversity of the post-Cambrian brachiopods and bivalves using fossil occurrence data from the Paleobiology Database (PBDB). This analysis shows the long-term pattern of their diversification history and highlights the importance of the Permian–Jurassic interval in assessing the brachiopod-bivalve switch. We then exhaustively revise the taxonomy and stratigraphy of the global fossil record of Permian–Jurassic brachiopods and bivalves and use these bespoke datasets to examine the dynamics of the brachiopod-bivalve switch at high temporal resolution, with emphasis on differences between ecological guilds and geographical regions. Finally, we use a multivariate birth-death model (PyRateMBD[38]) to evaluate the potential factors driving the extinction and recovery of brachiopods and bivalves across the PTME.

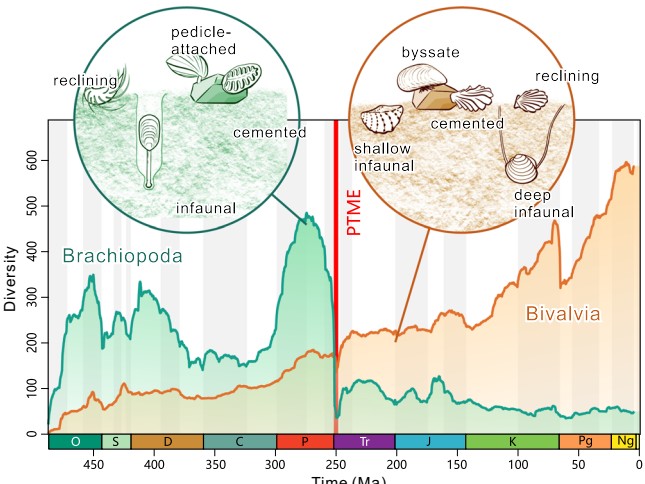

**Fig. 1 | Diversities of brachiopods and bivalves over the past ~487 Myr, showing the brachiopod-bivalve switch near the Permian-Triassic boundary.** Data from Fig. 2. Brachiopods were diverse in the Palaeozoic but were severely affected by the Permian-Triassic mass extinction (PTME), while bivalve diversity gradually increased, showing the brachiopod-bivalve switch near the Permian-Triassic boundary. The left circle shows typical lifestyles of Permian brachiopods. The right circle illustrates the major types of Triassic and Jurassic bivalves. O Ordovician, S Silurian, D Devonian, C Carboniferous, P Permian, Tr Triassic, J Jurassic, K Cretaceous, Pg Paleogene, Ng Neogene.

## Result and discussion

### Diversification dynamics of brachiopods versus bivalves

The Bayesian analyses of post-Cambrian datasets show that the large-scale diversification dynamic patterns of brachiopods and bivalves were commonly shaped by major biotic and environmental events, such as the Great Ordovician Biodiversification Event[39], the 'Big Five' mass extinctions[6,7] and their subsequent recovery phases (Fig. 2). A closer examination of the diversification rates, however, indicates very different patterns of the two clades before and after the Triassic-Jurassic boundary (i.e., largely similar trends from the Ordovician to Triassic vs. largely divergent trends from the Jurassic to recent). Despite different volatilities (which may have been resulted from different preservation rates), brachiopods and bivalves share a gradually decreasing trend in extinction and origination rates during the early to middle Palaeozoic, typical of rates of all marine animals[6,12,40]. In the Jurassic, brachiopods experienced frequent extinction and origination events but the rates of bivalves were largely stable. After the Jurassic, brachiopods possessed almost stable extinction and origination rates except for a minor surge in extinction rate across the Cretaceous-Paleogene (K-Pg) boundary (Fig. 2d, e), while bivalves had much more volatile extinction and originating rates (Fig. 2a, b). The flat rates of the post-Jurassic brachiopods are unlikely to have been caused by their low diversity because the confidence interval is rather narrow, and PyRate has the ability to detect significant rate shifts in such a situation[36,37]. Overall, the Bayesian analyses show comparable trends of diversification dynamics of pre-Jurassic brachiopods and bivalves, but less similar trends for post-Triassic taxa. A similar pattern is also found in analyses of the same datasets using the traditional method, i.e., the per-capita rate[41] (Supplementary Figs. 39, 40, 43–48). The correlation between brachiopod and bivalve extinction rates is more prominent than that of their origination rates, indicating that their extinctions were generally caused by similar environmental events[42]. The newly calculated diversities of post-Cambrian brachiopods and bivalves are largely comparable with those previously published[5,19,43–45] and clearly demonstrate the brachiopod-bivalve switch across the PTME (Figs. 1, 2c, f).

Given the different evolutionary trends of brachiopods and bivalves before and after the Triassic-Jurassic boundary and the most pronounced diversity switch of brachiopods and bivalves occurring over the Permian–Triassic transition (Fig. 1), the newly revised datasets of Permian–Jurassic brachiopods and bivalves were analysed in detail. To enhance temporal resolution, the revised Permian–Jurassic datasets consist of stratigraphical occurrences that were assigned to the substage level when possible. The general trends derived from the emended datasets (Fig. 3) remain similar to those derived from the uncorrected data in the PBDB (Fig. 2) and largely agree with the rates estimated for discrete time bins[46] (Supplementary Figs. 41, 42), but show more detailed fluctuations of rates. From the Early to Middle Permian, both brachiopods and bivalves showed decreasing origination rates, while brachiopod extinction rate exhibited a more pulsed, episodic pattern (Fig. 3a, d). The PTME severely impacted both groups, but the extinction rate of brachiopods was almost double that of bivalves (Fig. 3a, d). As a result, brachiopod diversity declined more sharply than that of bivalves across the PTME (Fig. 3a–f).

After the PTME, both groups underwent rapid diversifications in the first 10 Myr of the Triassic. More detailed comparisons indicate that bivalves recovered immediately after the PTME, while brachiopods rebounded ~2 Myr later, in the Olenekian (Fig. 3a, d). In the Ladinian, both groups displayed decreasing origination rates, followed by a pronounced surge in origination rate at the beginning of Late Triassic Period (Fig. 3a, d). Their origination and extinction rates then remained relatively consistent until the end of the Triassic, despite weak fluctuations, where both groups exhibited concordant spikes in origination and extinction rates during the Triassic-Jurassic mass extinction (TJME; Fig. 3a, d).

Jurassic bivalves and brachiopods displayed markedly different extinction and origination rates. Bivalves showed no shifts in origination rate, but their extinction rate curve peaked sharply at the Middle-Late Jurassic boundary and again at the end of the Jurassic (Fig. 3a). In contrast, brachiopods experienced a permanent increase in extinction rate during the Toarcian oceanic anoxic event and a further, albeit temporary spike near the Middle-Late Jurassic boundary (Fig. 3d). Although suffering high extinction rate, brachiopods still achieved a high origination rate in Middle Jurassic (Fig. 3d).

The decline of brachiopods and rise of bivalves coincide with the PTME[5]. However, brachiopods suffered more, with several orders wiped out[44,47], whereas the main groups of bivalves were not eliminated[45]. Consequently, it is somewhat inappropriate to compare the diversity of all Palaeozoic brachiopods with that of all Mesozoic bivalves. We therefore subdivided brachiopods into two major groups: the orders that went extinct in the PTME and the orders that survived or originated after the PTME (referred hereafter as PT$_e$ and PT$_s$ groups, respectively; see Methods for the detailed order names and assignments). Our results reveal that PT$_s$ brachiopods did not show a strong trend of long-term diversity decline despite a sharp drop across PTME (Fig. 3i). Instead, their diversity curve frequently fluctuated, and by the Middle–Late Jurassic they returned to levels previously displayed in the Permian (Fig. 3i). Clearly, the superficial decline of the entire brachiopod phylum after the PTME is likely caused by the extinction of some diverse Palaeozoic groups[5], rather than a failure by the PT$_s$ brachiopods to re-diversify. PT$_s$ brachiopods did re-diversify in Middle–Late Triassic, and Early and Middle

Jurassic, although their diversity remained lower than that of bivalves (Fig. 1).

Overall, the brachiopod-bivalve diversity replacement was neither the 'double-wedge' pattern[48] (i.e., brachiopod diversity gradually declined, while bivalve diversity increased), nor the 'mass extinction' pattern[48] (i.e., bivalves could only diversify after the extinction of brachiopods) (Fig. 1). Based on assumptions of ecological saturation and competition between brachiopods and bivalves, Sepkoski[19] used the coupled logistic model to explain the observed diversity pattern of the two clades. That model assumes that the increase of bivalve diversity decreases the net (i.e., origination minus extinction) diversification rate of brachiopods, and the solution to the model requires a decline in brachiopod diversity in the middle−late Palaeozoic and a decline in net diversification rate of bivalves when approaching the present[19]. Obviously, the newly calculated diversification rate patterns do not support it, nor do the high brachiopod diversity in the Permian (the peak is also prominent after sampling standardisation[49,50]) or the rapid and non-stop increase in Cenozoic bivalve diversity[43,51]. Conversely, Steele-Petrović[17] hypothesised that the Mesozoic replacement of bivalves was achieved by a series of extinction events. Because of greater abilities of bivalves to resist extinction, to colonise, and to expand distribution, they rapidly invaded vacant habitats left by brachiopods after mass extinctions, and then prevented brachiopods from occupying these ecospaces, essentially as the 'incumbent replacement model'[52]. The superiority of bivalves as Steele-Petrović argued is shown by the fossil record: they rapidly 'took over the world' after the PTME, and had a much higher abundance than brachiopods in the

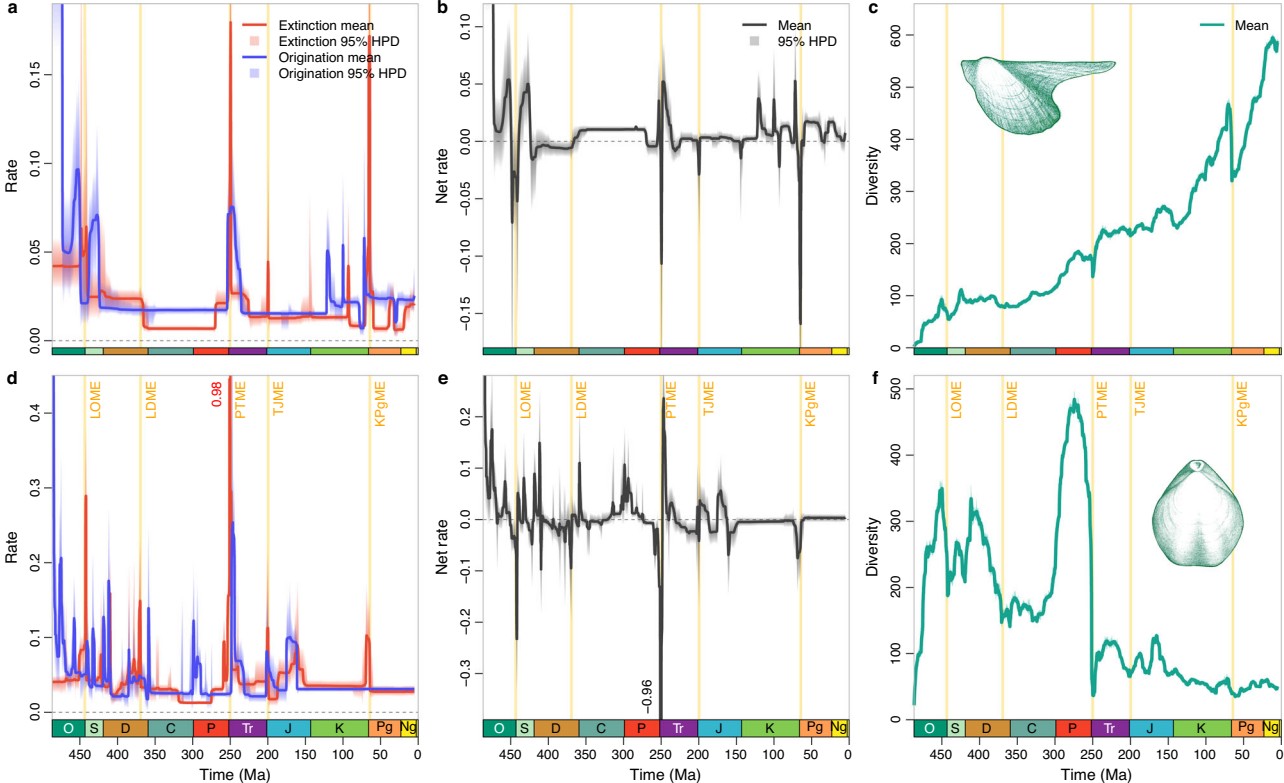

**Fig. 2 | Estimated diversification dynamics and diversities of post-Cambrian bivalves and brachiopods.** Trends in bivalves (**a**–**c**) and brachiopods (**d**–**f**), with shaded areas in **a, b, d, e** indicate 95% highest posterior density (HPD) intervals; those in **c** and **f** indicate estimations of different replications that incorporate age uncertainties of fossil occurrences. The two clades displayed comparable large-scale trends of diversification dynamics, especially before the Jurassic, viz., generally and gradually decreasing origination and extinction rates from the Ordovician to Carboniferous, and high extinction rates in the 'Big Five' mass extinctions: Late Ordovician mass extinction (LOME), Late Devonian mass extinction (LDME), Permian-Triassic mass extinction (PTME), Triassic-Jurassic mass extinction (TJME), and Cretaceous-Paleogene mass extinction (KPgME). The exceptionally high origination rate in the earliest Ordovician may be biased by the edge-effect because Cambrian occurrences were not included in the analysis. O Ordovician, S Silurian, D Devonian, C Carboniferous, P Permian, Tr Triassic, J Jurassic, K Cretaceous, Pg Paleogene, Ng Neogene.

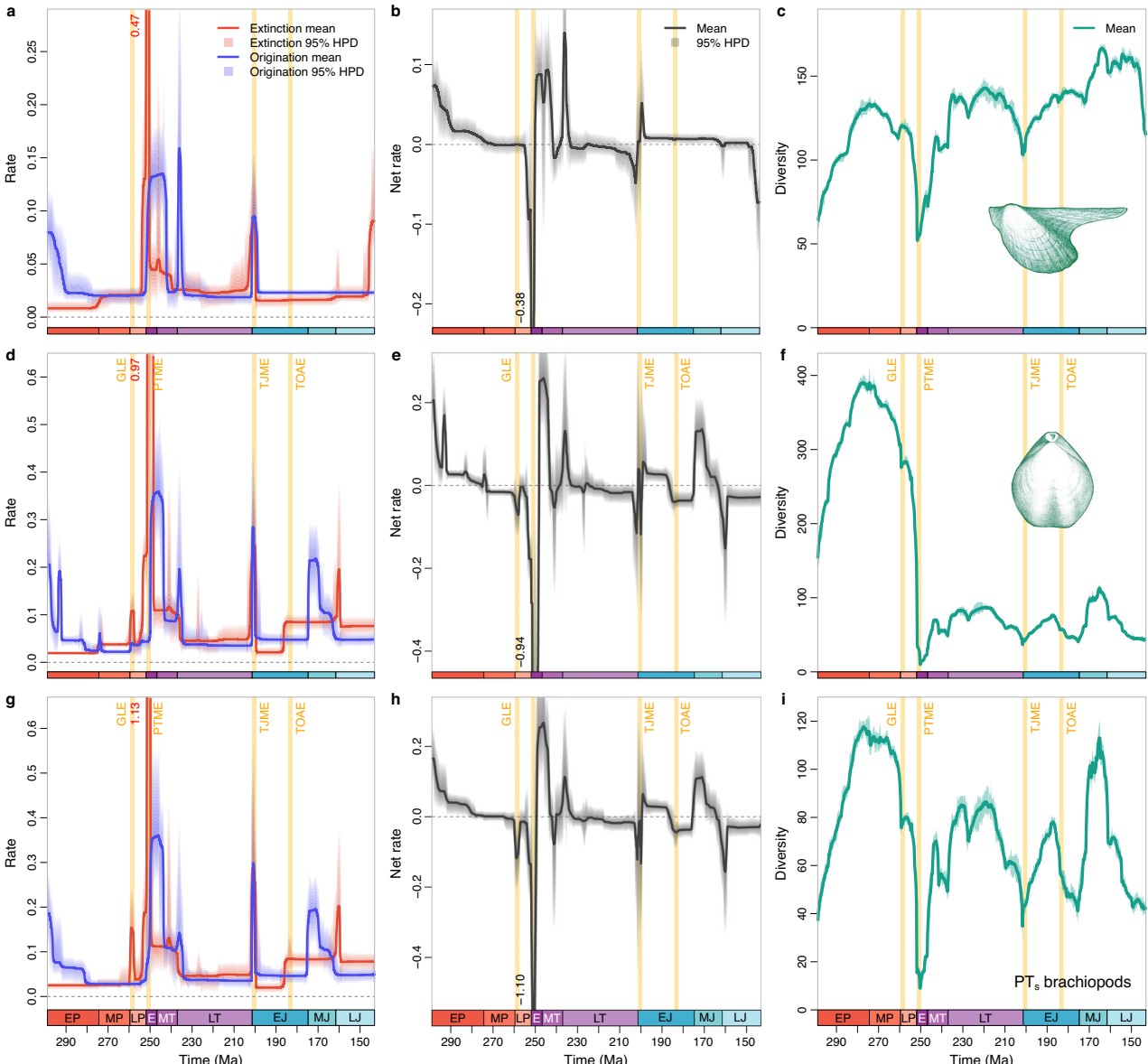

**Fig. 3 | Estimated diversification dynamics and diversities of bivalves and brachiopods during the Permian-Jurassic.** The shaded areas in **a**, **b**, **d**, **e**, **g**, **h** indicate 95% highest posterior density (HPD) intervals; those in **c**, **f**, and **i** indicate estimations of different replications that incorporate age uncertainties of fossil occurrences. Both bivalves (**a–c**) and brachiopods (**d–f**) were greatly affected by the Permian-Triassic mass extinction (PTME) and Triassic-Jurassic mass extinction (TJME), and both experienced high origination rates in the Early–early Middle Triassic, Carnian, and earliest Jurassic. Brachiopods also showed elevated extinction rate in the Guadalupian-Lopingian extinction (GLE) and Toarcian oceanic anoxic event (TOAE). Diversity of the Brachiopoda dramatically declined after the PTME and never recovered to the pre-extinction level. However, brachiopod orders that survived or originated after the PTME (PT$_s$ brachiopods, **g–i**) did not show an apparent decline in diversity in Triassic–Jurassic. EP Early Permian or Cisuralian, MP Middle Permian or Guadalupian, LP Late Permian or Lopingian, E Early Triassic, MT Middle Triassic, LT Late Triassic, EJ Early Jurassic, MJ Middle Jurassic, LJ Late Jurassic.

earliest Triassic[21,23,25]. However, the presence of bivalves did not prevent brachiopods from recovering in the Olenekian and Anisian as expected by the model (Fig. 3). In addition, the brachiopod radiation in the Middle Jurassic also indicates that brachiopod diversification was not prohibited by the bivalves. All the evidence suggests that brachiopods and bivalves had their own evolutionary paths. Neither the 'passive replacement model'[48] nor the 'active displacement model'[19,48] can sufficiently explain their diversification history.

### Diversification dynamics between ecological groups
For brachiopods, there is a heterogeneity of rates between the different lifestyles (cemented, reclining, pedicle-attached, infaunal) (Supplementary Figs. 5–9). Infaunal brachiopods did not show significant

rate shifts, but the other three groups displayed similar trends to one another in the Permian. Cemented and reclining taxa suffered a higher extinction rate during the PTME than pedicle-attached taxa, as most of the former belong to the PT$_e$ group. Although a new cemented clade, Thecideida, arose in the Triassic, its fossil record is sparse and thus did not have a strong impact on the rates.

Bivalves have a greater number of ecological lifestyles than brachiopods[53]. According to their mode of life and relative position of the animal and sediment, we categorised them into five groups: cemented, reclining, epibyssate (including semi-infaunal endobyssate; see Methods), shallow infaunal, and deep infaunal (Fig. 4, Supplementary Figs. 12–18). The large-scale patterns show that both epifaunal and infaunal bivalves possessed elevated extinction rates in the PTME

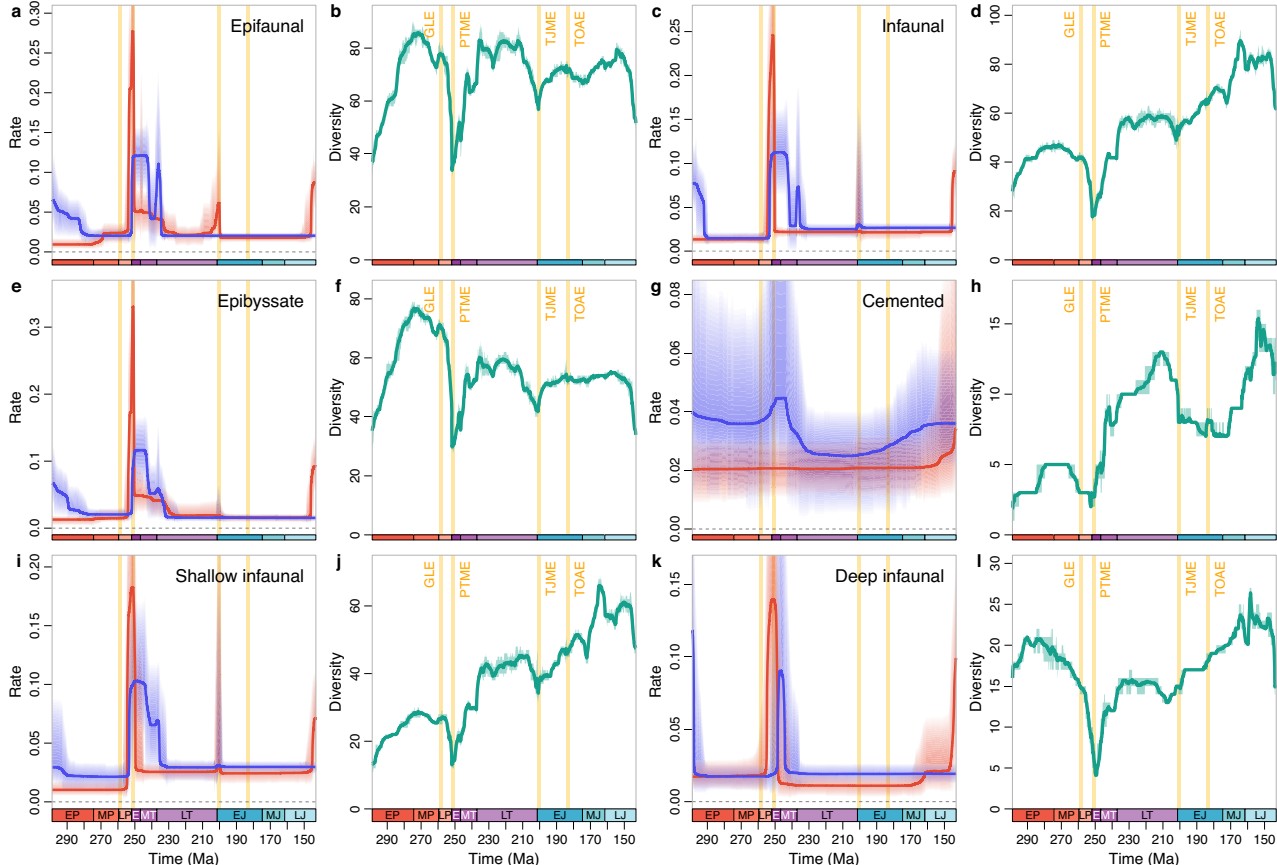

**Fig. 4 | Origination and extinction rates, and diversity of bivalves with different ecological lifestyles.** The shaded areas in **a, c, e, g, i, k** indicate 95% highest posterior density (HPD) intervals; those in **b, d, f, h, j, i** indicate estimations of different replications that incorporate age uncertainties of fossil occurrences. Epifaunal bivalves (**a, b**) includes three groups: epibyssate (including semi-infaunal endobyssate) (**e, f**), cemented (**g, h**) and reclining (Supplementary Fig. 16). Infaunal bivalves (**c, d**) includes two groups: shallow infaunal (**i, j**) and deep infaunal (**k, l**). Most of these groups displayed a high extinction rate in the PTME and high origination rate in the Early–early Late Triassic. In most of the Late Triassic–Jurassic, epifaunal bivalves had a net diversification (origination minus extinction) rate close to zero (**a**, Supplementary Fig. 12), lower than that of infaunal bivalves (**c**, Supplementary Fig. 13), resulting in the relatively stable diversity of epifaunal group and increased diversity of infaunal group. Abbreviations as in Fig. 3.

and TJME, decreasing origination rates through the Permian, and elevated origination rates through Early Triassic to Carnian (Fig. 4a, c). Both groups rebounded rapidly from the PTME, with similar origination rates. Nevertheless, the extinction rate of epifaunal bivalves remained relatively high in the Triassic like brachiopods (Fig. 2d, g), while that of infaunal bivalves sharply dropped after the PTME, and remained unchanged except at the ends of the Triassic and Jurassic (Fig. 4a, c). After the Norian, epifaunal taxa maintained a constant origination rate (mean $\lambda_{\text{Middle Jurassic}} = 0.020$), while infaunal taxa showed a slightly higher absolute value (mean $\lambda_{\text{Middle Jurassic}} = 0.027$) coupled with a further origination episode after the TJME.

In the Triassic, the extinction rate of epifaunal bivalves was generally higher than that of infaunal bivalves, as demonstrated previously[46,54]. In the Jurassic, the origination rate of epifaunal bivalves was low or very slightly higher than their extinction rate, while infaunal bivalves showed higher origination rates than extinction rates (Fig. 4a, c). The differences in net diversification rate between the two ecological categories therefore underpin their respective evolutionary trends in diversity: the relative proportion of infaunal bivalves increased gradually while epifaunal taxa correspondingly decreased (Supplementary Fig. 27), supporting an infaunalization process observed by previous authors[46,54,55] which continued to the present[53]. This infaunalization has been linked to an early onset of the Mesozoic marine revolution[13] by some authors[46,55], but the lack of abundant predators in the Triassic suggests an alternative interpretation[54].

Although some predators presumably originated in the Late Triassic to Jurassic[46], according to our results, very few shifts in epifaunal and infaunal origination rate occurred during this period. Some groups developed novel adaptations that may have been helpful in the arms race between predator and prey[56,57], but these innovations did not cause either group to diversify rapidly and substantially. Besides, shallow infaunal taxa, the key group of infaunal bivalves, had a higher extinction rate than epibyssate ones (mean $\mu_{\text{Middle Jurassic}} = 0.025$ versus mean $\mu_{\text{Middle Jurassic}} = 0.016$, respectively; Fig. 4e, f, i, j), indicating that an infaunal lifestyle did not protect them from extinction. Alternatively, the slightly higher extinction rate and even higher net diversification rate of infaunal bivalves may exhibit their intrinsic ecological success independent of any response to predation pressure. As a result, our results demonstrate that Triassic–Jurassic bivalves show no obvious sign of the escalation-driven diversification that left a mark on their evolutionary rates. Further studies on smaller temporal and spatial scales may illuminate when and how predators affected the evolution of bivalves[58] (and brachiopods).

### Diversification dynamics between geographic regions

Diversification rates and diversities of brachiopods and bivalves show distinct regional heterogeneity, highlighting the importance of biogeographic nuance in broader patterns of diversity[59] (Supplementary Figs. 19–26). Except for in south-western Tethys, brachiopods and bivalves in all other regions displayed an increase in net diversification

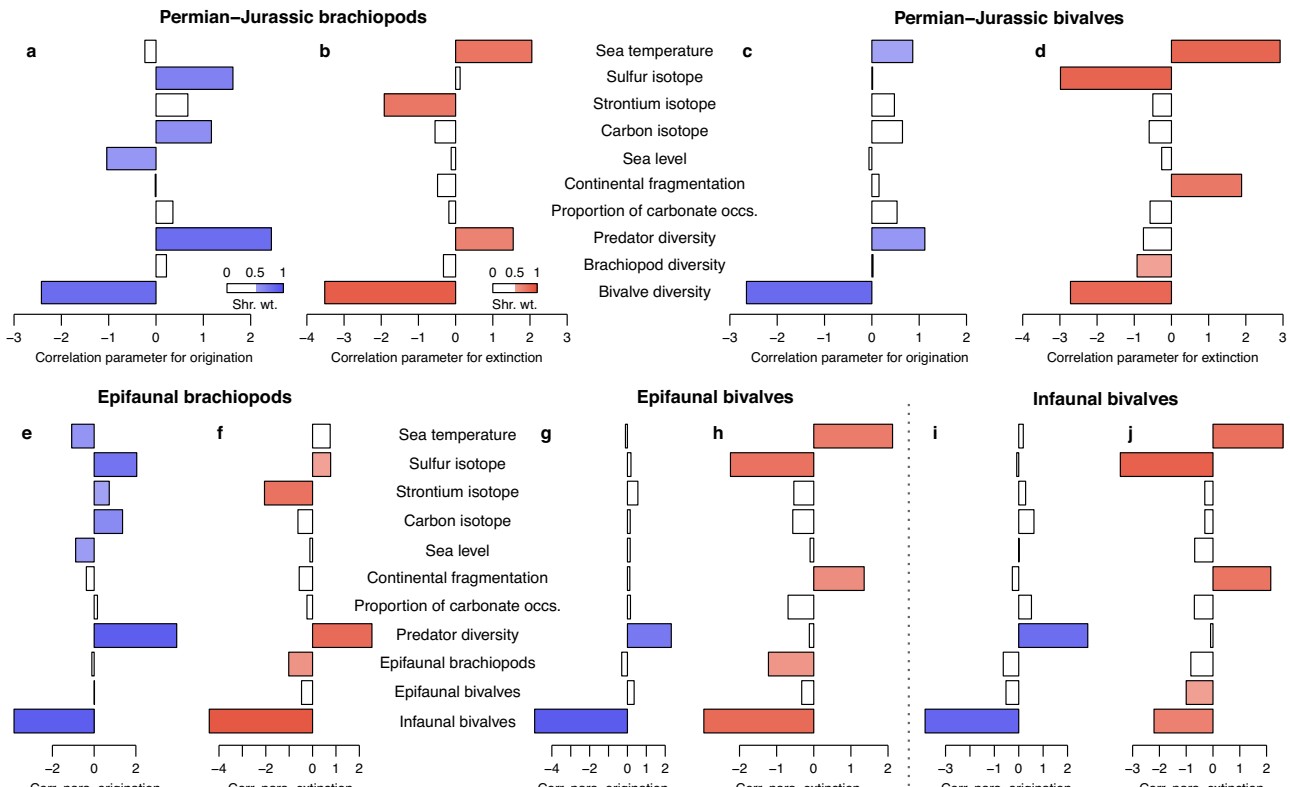

**Fig. 5 | Estimated correlation parameters on origination and extinction rates with abiotic and biotic factors.** Bivalves are either considered as a whole (**a**–**d**) or categorised into two groups, epifaunal bivalves and infaunal bivalves (**e**–**j**). All results in the figure are based on the long-time window (Permian–Jurassic) analysis. A filled bar indicates the relationship is significant (shrinkage weight [shr. wt.] $w > 0.5$). Corr. Correlation, para. parameter, occs. occurrences.

rate after the PTME. Both north-western Tethys and eastern Tethys possess the most abundant fossil occurrences and bear a higher similarity in rates to the global trends, corroborating previous findings that the signal from the 'global' fossil record may instead be disproportionately driven by data from one or two well-sampled regional records[60]. In these two regions, brachiopods showed a slightly delayed increase in origination rate after the PTME compared to bivalves. Although the multivariate birth-death (MBD) model was not applied to regional data to explicitly infer the drivers of origination and extinction, the estimated rates indicate that brachiopod origination was neither suppressed by bivalve diversification, nor stimulated by their extinction. Further, bivalves did not necessarily have a higher net diversification rate than brachiopods.

## The role of competition

We conducted multivariate birth-death (MBD) analyses on the diversification dynamics for the Permian–Jurassic period as a long-time window analysis, which is further subdivided into four smaller time windows: Asselian–Wordian, Capitanian–Ladinian, Carnian–Toarcian, and Aalenian–Tithonian. Both long- and short-time window analyses have advantages and disadvantages. Long-time window analysis is more useful in revealing factors that contribute to large-scale and long-term processes that shape the basic pattern of a clade's diversification history, but if the effects of predictors are highly variable through time, the combined effects of predictors may not explain the observed diversification rates[61]. Short window analysis reveals time-varying relationships[61,62], however, it risks over-parameterisation, and the noise of a largely stable factor may be magnified in a short-time window analysis, affecting the evaluation of other factors.

We first performed the MBD analysis on brachiopods and all bivalves, and found that many factors correlated with their diversification dynamics (Fig. 5a–d). Of these, their diversities are one of the

most outstanding factors shaping their rates. If brachiopods and bivalves competed with each other, increased diversity of one clade should reduce the origination rate and/or increase the extinction rate (= reduce the net diversification rate) of the other[19]. Bivalve diversities in some intervals (Permian–Jurassic, Capitanian–Ladinian, and Carnian–Toarcian) show strong negative correlations with origination rates of brachiopods (Fig. 5a; Supplementary Tables 1, 3, 4), suggesting competitive pressure as conventionally argued. However, such an interpretation requires caution as the negative relationship between diversity and origination rate also affected bivalves themselves (Fig. 5c). In fact, this negative correlation is largely caused by the rebounded origination after the extinction when diversity is low[43,51,63,64]. For brachiopods and bivalves, this 'diversity-dependent' pattern is only prominent in the aftermaths of mass extinctions. During 'normal' times without mass extinction (e.g. Asselian–Wordian, Aalenian–Tithonian), this 'diversity-dependence' was not uncovered, and brachiopods were not 'suppressed' by bivalves (Supplementary Tables 2, 5, 7, 10). The reason why bivalve diversity rather than brachiopod diversity played a role (Fig. 5) is simply because the bivalve diversity trajectory (Fig. 3c) is closer to the shape of origination rate curves (Fig. 3a, d). Bivalve diversity rapidly decreased then increased across the PTME and TJME. As a result, the sharp declines in diversity correspond well to the spikes of extinction and origination rates (Fig. 3). Accordingly, it is likely that diversity loss during mass extinction stimulated the increase in origination rates of both brachiopods and bivalves, but brachiopods were not necessarily competitors of bivalves (i.e., the common cause effect[65]).

Of the various bivalve life-modes, epifaunal taxa have larger overlaps with brachiopods (which are mostly epifaunal) in food and habitat than infaunal bivalves. Therefore, epifaunal bivalves should be the direct competitor of brachiopods if bivalves and brachiopods do have this relationship[19,30]. Infaunal bivalves, in contrast, have more

diverse feeding behaviours and occupy different niches from brachiopods. Although the activities of infaunal bivalves may disrupt substrates and thus affect occupation by brachiopods[66,67], it is a quite different mechanism from direct competition. For this reason, the MBD analysis was also conducted for the three ecological groups, epifaunal brachiopods, epifaunal and infaunal bivalves. Similar to the result described above (Fig. 5a–d), the new results also indicate that 'diversity-dependence' is an important mechanism that regulates their origination rates (Fig. 5e–j): the infaunal bivalve diversity strongly and negatively correlated with origination rates of all three groups. This 'diversity-dependence', or the elevated origination rate after mass extinction, is a ubiquitous phenomenon across multiple marine clades[43,51,63,64,68], which has been attributed to weakened competition[43], extinction of predators[51], or others. Regardless of specific ecological meaning, these correlations again suggest that the long-term (i.e., Permian–Jurassic) diversification dynamics of the three groups were largely governed by comparable factors, and bivalves (or specifically, epifaunal bivalves) were not competitors of brachiopods over macroevolutionary scales.

Liow et al.[20] estimated diversification and sampling rates of post-Cambrian brachiopods and bivalves using an alternative method, and analysed the 'causative' or 'correlative' relationship of these rates based on stochastic differential equations. They detected a 'causative' link between bivalve extinction and brachiopod origination[20] and concluded that brachiopods and bivalves are more than 'ships that pass in the night'[5]. Nevertheless, similar to our results (especially the per-capita rates estimated in discrete time bins as their method; Supplementary Figs. 39, 40, 43–45), they found that the diversification rates of brachiopods and bivalves correlated with one another[20]. As a result, brachiopod extinction also 'caused' brachiopod origination[20] and therefore, it is difficult to tell whether it was bivalves, or other factors affecting both clades (i.e., extinction events) that triggered the origination of brachiopods. To deal with this problem, Reitan and Liow[27] re-analysed the data in multivariate models. The best model confirmed their previous view that bivalve extinction 'caused' brachiopod origination, but some relationships in this model[27], such as brachiopod origination causing bivalve origination, bivalve extinction causing brachiopod extinction, do not fit well with ecological theories (although they interpreted these relationships as reflecting similar origination and extinction rates of brachiopods and bivalves[27]).

The absence of competition on macroevolutionary scales does not mean that brachiopod-bivalve competition cannot occur in small temporal and spatial units. Their similar ecologies would suggest that such competition could occur[17]. For example, through cage experiments, Thayer[18] observed that mussels can increase the mortality of brachiopods especially when predators are absent. In the fossil record, intermittently segregated distribution of brachiopods and bivalves was also observed in some regions[69]. However, both modern and fossil assemblages suggest that such competitive exclusion rarely occurs when predators and other disturbances are present[31,70]. Besides, when brachiopods and bivalves coexist, the physiological 'superiority' of bivalves does not guarantee their dominance. In today's ocean, there are some regions where brachiopods are not affected by bivalves, but instead are much more abundant than bivalves[71,72]. The opportunistic settlement of these animals means the colonisation of a species is largely a matter of chance; the result is many species can coexist, regardless of their competitive abilities[31,72].

Extrapolating microevolutionary dynamics to macroevolutionary time scales is potentially problematic[19,48,73]. Nevertheless, for competition-driven macroevolutionary patterns, it is intuitively assumed that competition could decrease population size or growth rates of the inferior clade, raising its extinction probability[73]. If the competition between brachiopods and bivalves was intense and pervasive, then should have affected their diversification rates, which is not observed in our results. Instead, while competition exclusion

driven by bivalves could have happened sporadically, the cumulative evolutionary pressure of such interactions was negligible[29].

## The role of top-down drivers

Many external factors correlated with the diversification dynamics of brachiopods and bivalves. For origination rates, the predator diversity and the 'diversity-dependence' discussed above had the highest absolute value of coefficients and contributed greatly to the long-term pattern. For extinction rates, sulfur and strontium isotope excursions were usually negatively correlated with them, while sea temperature and continental fragmentation generally had a positive correlation parameter (Fig. 5).

A comparison of brachiopod and bivalve responses to predictors over various time windows shows that almost no environmental predictors displayed a clear positive correlation with the bivalve diversification but a negative one with brachiopods (i.e., the selection of bivalves against brachiopods). Sea temperature is an exception. In the long-time window analysis (Permian–Jurassic), brachiopod and bivalve origination rates showed different responses to temperature (Fig. 5). When focusing on the critical time of the brachiopod-bivalve switch (i.e., Capitanian–Ladinian; Supplementary Tables 3, 8, 13, 18, 23), this difference is much more significant as temperature had a very high negative coefficient with brachiopod origination rate, while the correlation coefficient with bivalve origination rate is positive (although insignificant). This indicates that high temperatures in the Early–Middle Triassic possibly inhibited the origination rate of brachiopods but did not affect (or perhaps even stimulated) the bivalve origination rate. This interpretation is supported by the timing of recovery of the two groups. Seawater temperature increased rapidly to exceptionally high values after the PTME[74] and in this hot environment bivalves diversified with a high origination rate (Fig. 3). But for brachiopods, even though rare new Mesozoic-type taxa originated in the Induan, their main diversification phase occurred in the Olenekian to Anisian when temperatures cooled[74-76] (Fig. 3). Greater bivalve tolerance to hyperthermal conditions has also been noted in the Toarcian oceanic anoxic event when brachiopods experienced a more prominent extinction and reduction in body size than bivalves[77-79]. Among modern taxa, although data are sparse, brachiopods seem to be more sensitive to warming than bivalves[80]. Together with a higher extinction rate of brachiopods at the PTME, these results support the hypothesis that physiological differences between brachiopods and bivalves caused the brachiopod-bivalve switch[21,23,81,82].

In contrast to the 'competition' hypothesis, Stanley[30,31] suggested that the diversification of advanced predators caused the decline of brachiopods, but our analyses do not fully support this scenario. Predator diversity positively correlated with the extinction rate of Permian–Jurassic brachiopods, which seems to support Stanley's hypothesis. However, the increasing predatory diversity was also accompanied by an increase in brachiopod origination rate. More importantly, the Cenozoic diversification of predators did not cause a substantial increase in extinction rate or decrease in origination rate for brachiopods because their net rates were even slightly higher in the Cenozoic than Cretaceous (Fig. 2e). This is possibly because brachiopods experienced intensified predation before the Cenozoic and had become marginal components of benthic ecosystems, so they were no longer subject to predators. Alternatively, predation may not be the key force driving the decline of brachiopods, which is reinforced by the observation that brachiopods are typically not the first targets of predators[83,84].

The decline of brachiopods was also attributed to their limited ability to expand their distribution after the PTME[17,32,33]. It has been shown that in the Triassic and Jurassic, the longitudinal distribution of brachiopod genera was restricted[33], possibly resulting in a higher extinction risk[85,86]. However, our MBD analyses suggest that continental fragmentation seems not to have had a noticeable effect on

brachiopods (Fig. 5). Rather, it correlated positively with the bivalve extinction rate. This relationship is difficult to explain because it was not seen in all short-time window analyses, and further studies are needed to test it. For brachiopods, restricted geographical distribution did not always negatively affect their diversity, as narrowly distributed taxa also tend to have a higher origination rate[85]. For instance, many Anisian brachiopods were endemic taxa that did not persist into the Late Triassic[87], but they contributed substantially to the Middle Triassic recovery of brachiopods.

### Study limitations

Although we modified some data in the PBDB, there are always problems with taxonomy of fossils, which is inevitable in any palaeontological studies[88]. In some cases, the hinge morphology of bivalves and internal characters of brachiopods are not well preserved, preventing precise identification. Considering the long temporal span and large size of the dataset, these taxonomic issues should not severely affect the estimation of rates[88]. The analyses of a particular group or a region, nevertheless, might be strongly influenced, especially when the dataset is small.

A further limitation is that we were not able to include all possible environmental proxies, such as seawater oxygen content and pH values[89,90] because of the lack of long-term and continuous records. The hypothesis that the brachiopod decline was caused by habitat loss[17] was not tested because it is difficult to assess the size of habitats preferred by brachiopods. Further, for the geochemical proxies analysed, the temporal density of datum points is highly variable. 'Hot' intervals like the Permian-Triassic boundary are well documented, while some intervals such as the Late Triassic are poorly studied, hindering the discovery of environmental events and potential relationships.

The analyses of regional datasets extract the spatially heterogenous rates concealed by the global signal, but they could be greatly biased by regional sedimentation, tectonic and other factors due to the relatively small size of the spatial windows[60]. In addition, the current curves of some geochemical proxies are composed of data from all over the world, which can be spatially heterogeneous. As a result, attempting regional MBD analyses over long time scales is challenging.

Other confounding factors, such as the precise ages of fossil occurrences and uneven sampling are unavoidable, although these uncertainties and heterogeneities are partially addressed by our Bayesian methodologies. Finally, concerns have been raised about whether diversity and diversification rates can be used as proxies for assessing biotic interactions in the fossil record (e.g., competition, predation-driven evolution) in place of the detailed data on individuals and populations available to modern ecological studies[91]. Resolving this question is beyond the scope of this paper, but our results nonetheless indicate that large-scale diversity patterns do not show any trace of competitive replacement between brachiopods and bivalves. Future studies on finer temporal and spatial scales (for both palaeontological and geochemical data) may resolve some of these problems and enable more precise tests of the hypotheses we present here.

### Concluding remarks

Although brachiopods and bivalves had different diversity trends, their diversification dynamics were of broadly similar patterns before the Jurassic. Through detailed study of Permian–Jurassic taxa, we confirmed that the well-known brachiopod-bivalve switch was caused by the PTME, which killed most brachiopods. In addition, the different responses of brachiopods and bivalves to high seawater temperature may have further accelerated this switch. Contrary to conventional wisdom, we find no signs of long-term competition between brachiopods and bivalves. In the Triassic–Jurassic, the surviving brachiopod lineages successfully re-diversified without being suppressed by the bivalves. The vast difference between bivalve and brachiopod

diversities today is largely due to great diversity loss of brachiopods in the PTME and rapid bivalve diversification in the Cretaceous and Cenozoic[51] which brachiopods failed to match. Consequently, the post-extinction (PTME) ecological replacement became evident mainly from the Late Jurassic onwards, 100 Myr after the PTME. Why the bivalve origination was rapid and unconstrained and what suppressed brachiopod origination after the Jurassic when origination contributed more to the diversity[92] may also help answer why brachiopods and bivalves are so different in diversity and abundance in the modern oceans. This study shows that brachiopods and bivalves had their own evolutionary paths. Their diversification histories were strongly controlled by rapid changes in environments (e.g., mass extinctions), and their differing physiological abilities to cope with these changes shaped their divergent fates.

## Methods

### Fossil occurrence data

To investigate the long-term patterns of bivalve and brachiopod diversification dynamics, we first investigated their post-Cambrian (Ordovician–Quaternary) fossil records. Cambrian records were not considered because bivalve fossil occurrences from this time are so rare. We then focused on their rates from the Permian to Jurassic, which is the critical interval when the brachiopod-bivalve switch took place[21]. All analyses were carried out at genus level.

Fossil occurrence data of brachiopods and bivalves were manually downloaded from the Paleobiology Database (PBDB) on 23/11/2022. Occurrences with ambiguous generic assignments were excluded, but those with an unambiguous genus name and an open species name (i.e., the species name is sp.) were retained. Terrestrial records (indicated by the *environment* field in the PBDB) were excluded. Uncertain records (primary generic resolution noted by *cf.*, *aff.*, *?*, *informal*, *ex. gr.*, "") with identical primary and accepted generic names were discarded, while those with different names (i.e., the species was reassigned to another genus) were retained. In addition, absolute ages (maximum and minimum ages) of the occurrences were updated based on ages in the Geological Time Scale 2020[93] using the *chrono_scale* function of the *fossilbrush* R package[94]. Next, occurrences with a high temporal uncertainty (>10 Myr) but not from an international stage were removed from the dataset. For example, a Guadalupian (-15 Myr uncertainty) occurrence was discarded, but a Norian record was kept although the latter has a -18 Myr uncertainty.

This cleaning procedure was first applied to the post-Cambrian datasets. No taxonomic modifications were applied to the datasets, aside from deletion of bracketed subgenus names from within genus names. After data cleaning, 179,030 and 153,011 occurrences were retained for the bivalve and brachiopod datasets, respectively, including a total of 2484 bivalve genera and 3427 brachiopod genera.

For the Permian–Jurassic analysis (-298.9–143.1 Ma), in addition to the above revisions, more detailed curations were made to generate a taxonomically and temporally more precise database. It should be noted that, although we focused on diversification dynamics from the Permian to the Jurassic, all Carboniferous–Cretaceous occurrences were kept during the revision stage; these were useful in analysing the ranges of genera (the *pacmacro_ranges* function; see below). Moreover, to ensure the rates were accurate near the boundaries of the studied interval (i.e., earliest Permian and latest Jurassic) and not affected by edge effects, the dataset analysed actually included the Kasimovian–Valanginian (-307.0–132.6 Ma) occurrences.

The additional revision process of the Permian–Jurassic dataset includes four main steps.

(1) In addition to the PBDB data, we added 1522 and 4538 occurrences of Permian–Jurassic brachiopods and bivalves, respectively. These occurrences were all reported from China and have not yet been included in the PBDB. The bivalve occurrences added are largely from the Triassic; the brachiopod occurrences added are mainly

from the Permian, and a few Jurassic brachiopod occurrences are also included. These records are important especially for the regional analyses, considering that Triassic bivalve records and the Jurassic brachiopod records are relatively sparse in eastern Tethys in the PBDB.

(2) The name of every genus in the dataset was examined. Names that are currently not used and synonyms of other taxa were revised. In contrast to simply deleting the subgenus name as we did for the post-Cambrian data, every subgenus was checked, and whether it should be upgraded to genus level was decided based on recent literature[95]. If it was regarded as a genus, the subgenus name was retained and the genus name was discarded; if not, only the genus name was retained.

(3) The duration of every genus was checked, and doubtful records were discarded. The duration of each genus was based on well-curated databases of taxon stratigraphic ranges such as *Treatise on Invertebrate Palaeontology*[96,97], Sepkoski's compendium[8] (accessed through the *fossilbrush* package), and recent literature[95]. The *pacmacro_ranges* function in the *fossilbrush* package was used to generate an additional, statistical set of stratigraphic references. This function uses the stratigraphic density of occurrences to highlight doubtful occurrences. We used the default settings of the function to examine the occurrences, and those falling outside the 90% density intervals of their taxon stratigraphic range were treated as doubtful. These doubtful occurrences were checked, and highly unreliable ones checked against well-curated databases were discarded. It is true that deleting these data but not emending them will unavoidably omit some useful records, but revising them all is infeasible considering the size of the dataset. The unrevised very old or poorly described records would have negatively affected estimation of diversification rates. Thus, removing these doubtful records is better than retaining them in the dataset.

(4) The ages of the occurrences were updated. Where possible, occurrences were revised to substage level based on lithostratigraphic information (i.e., formation, member) and recent literature that report the ages of the stratigraphical units[60]. For Permian records, each stage was divided equally into early and late parts. The absolute ages of the local Triassic and Jurassic stages of New Zealand were updated according to the correlation of biozones[98] and Geological Time Scale 2020[93].

After these revisions, 58,319 occurrences in 684 genera, and 71,987 occurrences in 1352 genera were left in the bivalve and brachiopod datasets, respectively.

The PTME caused the extinction of many Palaeozoic brachiopod orders[44]. However, this catastrophe had little effect on high-level clades of bivalves[45]. To compare the diversification patterns between the Mesozoic brachiopods and bivalves, the Permian–Jurassic brachiopods were then categorised into two groups. One group (PT$_e$) consists of brachiopod orders that went extinct in the PTME (including the Spiriferida, Productida, Orthotetida, Orthida, and Dictyonellida). The other group (PT$_s$) is composed of brachiopod orders that survived the PTME (including the Lingulida, Terebratulida, Athyridida, Rhynchonellida, Spiriferinida, and Craniida) and originated after this event (the Thecideida).

## Ecological categories of brachiopods and bivalves

All the Kasimovian–Valanginian brachiopods and bivalves were assigned to an ecological category according to their mode of life and relative position of the animal and sediment. Bivalve lifestyles were categorised into five groups: epibyssate (note: including semi-infaunal endobyssate here), cemented, reclining, shallow infaunal, and deep infaunal. Epibyssate bivalves usually live above the sediment-water surface (epifaunal), while endobyssate forms have parts (semi-

infaunal) or all (infaunal) of their shells buried in the sediments. Although some shell forms are typical of epibyssate or endobyssate groups, transitional forms in morphology between the two are also commonly present. Semi-infaunal endobyssate taxa have parts of their shell exposed to the water and are vulnerable to predation, and so were classified together with fully epifaunal elements, but fully infaunal endobyssate genera were placed in the infaunal group. The recliners have a much lower taxonomic consistency compared with other ecological groups. These include bivalves with various types of shell morphology: taxa with a thin and flat shell (e.g., *Bositra*), a thick and heavy shell (e.g., *Neomegalodon*), or a highly inequivalve shell (e.g., *Exogyra*). Some free-living forms that could possibly swim were present during the Permian–Jurassic, but they were very rare and not a big ecological group. We categorised these possible 'swimmers' into two groups based on the following criteria: if the byssus is retained in adults, it was classified in the epibyssate group; if the byssal notch is absent and the overall shell morphology is closer to the adept swimmers, it was classified in the reclining group[99]. The epibyssate, cemented, and reclining taxa belong to the epifaunal group. The infaunal bivalves include shallow and deep infaunal groups. Shallow infaunal taxa are usually nonsiphonate; they live close to the water column and are easily exhumed by currents. Deep infaunal taxa are often siphonate and have a greater chance of avoiding predation. Borers that live in hard substrates were classified in the deep infaunal group given that they have a stronger resilience to waves and predators. The assignment to eco-group was based on the lifestyle of the adults inferred from their shell morphology, following Stanley[99–101], Ros-Franch et al.[95], Mondal and Harries[53], and other literature. Since a genus may have more than one mode of life, the major lifestyle of the genus was selected. A question mark (fewer than 1% of taxa) was given if the life strategy is poorly known or highly variable.

Unlike that of bivalves, the life strategy of brachiopods is mostly inferred because many fossil brachiopods do not have living analogues. The ecology of brachiopods given here was categorised entirely based on taxonomic classification rather than detailed morphology-function analysis. We categorised the brachiopods into four ecological types: pedicle-attached, cemented, reclining, and infaunal. The first three were clustered as the epifaunal group. Pedicle-attaching type is the most common lifestyle among the Rhynchonelliformea; we assigned all the orthides, athyridides, rhynchonellides, spiriferides, spiriferinides, and terebratulides to this ecological category. The cemented type includes those attached by the shell surface and/or with spines to support the body, such as the orthotetides, craniides, strophalosiidines, and lyttoniidines. The reclining type consists of the chonetidines and productidines. Although all have a concavoconvex shell, the strophalosiidines usually possess a cicatrix near the ventral umbo and rhizoid spines anchoring them to other objects, while the adult chonetidines and productidines normally lack these features, but instead develop long spines near the hingeline or on the body to stabilise them on the substrate[102]. The infaunal type consists only of the deep infaunal lingulides. There must be exceptions among these ecological types. For example, the pedicles of some rhynchonellides, terebratulides, and spiriferides may lose their function when the shells grow large, and therefore, the animal turns into a recliner. These exceptions were not considered in this study.

## Estimation of diversification rate

Previous studies[5,19] usually used taxonomic diversity to investigate the brachiopod-bivalve switch. However, diversity arises from the interplay of origination and extinction processes and the same diversity pattern can be generated by completely different diversification dynamics[37]. For example, an increase in diversity may have resulted from elevated origination rate or decreased extinction rate or both, and they correspond to different mechanisms. To investigate the drivers of diversification, separating origination and extinction processes

is essential. Diversification dynamics of all datasets were analysed in a Bayesian framework implemented in PyRate[35–37] (v.3). Incorporating both the fossil preservation process and the birth-death process, PyRate jointly estimates the preservation rates ($q$), the times of origination and extinction of each genus ($Ts$ and $Te$), and the origination and extinction rates ($\lambda$ and $\mu$). Differentiating from previous methods that estimate rate changes at discrete time points (usually stage boundaries), PyRate considers the diversification of a given group as a continuous process. Using the reversible-jump Markov Chain Monte Carlo (rjMCMC) algorithm, PyRate can explicitly estimate the number and temporal placement of statistically significant rate changes, minimising the risk of under- and over-parameterisation[37]. Analyses of simulated data indicate that PyRate generates reliable estimations of diversification rates under a variety of preservation models and outperforms traditional methods[36,37].

Because of our large datasets that include a great number of taxa (and parameters), it is challenging for the MCMC chain to converge. Therefore, we followed a two-step procedure[60] to estimate diversification rates for our data. First, we used PyRate with the Gibbs MCMC sampling algorithm[103] (in place of the default Metropolis-Hastings algorithm) to estimate the preservation-corrected $Ts$ and $Te$ of each taxon by sampling directly from their approximate posterior distribution. However, the rapid convergence achieved by this algorithm is at the cost of reduced resolution in estimated diversification rates[60]. To solve this problem, the second programme, LiteRate[104,105], was employed. LiteRate was developed to estimate the origination and extinction rates of large datasets. It does not consider the preservation process at all but has the same birth-death model as PyRate. Thus, preservation-corrected times ($Ts$s and $Te$s) generated by PyRate can be supplied to LiteRate to calculate high-resolution origination and extinction rates. In the end, all the preservation and birth-death parameters are properly estimated. Besides, $Ts$s and $Te$s estimated by PyRate were then used to run the multivariate birth-death analyses in PyRateMBD (see below).

To incorporate the age uncertainties of the fossil occurrences, ten replicates of the original dataset were generated. Occurrences of the same fossil assemblage/site, using the collection number as the indicator, were assigned a coeval age randomly taken from their stratigraphic age range. The analysis was done for every replication, and the parameters estimated by the ten replications were combined to plot the final result. For the preservation process, we used the -PPmodeltest function in PyRate to select the best fitting model from among a homogeneous Poisson process, a non-homogeneous Poisson process, and a time-variable Poisson process. The result supported the time-variable Poisson processes, which allows the preservation rate to vary as a piece-wise series of constant-rate time bins. The times of preservation rate shift were set to stage boundaries in the Permian–Jurassic analyses and to series boundaries in the post-Cambrian analyses. The preservation rates across different time bins were assigned a gamma distribution; the shape parameter was set to 1.5 and the rate parameter was set to 0 to allow PyRate to estimate the rate directly from the data (-pP 1.5 0).

Unlike previous studies[37,38,61,62], we did not model potential preservation rate heterogeneity between lineages (the Gamma model, -mG option) arising from differences in environment, ecological style, shell mineralogy, and other such biases. In theory, this heterogeneity is commonly present and should be included in analyses[37]. However, incorporation of the Gamma model increased the complexity of the overall model. When it is used along with the Gibbs model, for some datasets, the posterior values occasionally show sudden jumps even after a great number of iterations (Supplementary Fig. 28). For those replications that seemingly reached 'convergence' (the posterior samples are stable), the preservation rates may vary greatly between randomised replications, which indicates that the 'convergence' is spurious and that prohibitively many more generations would be

needed by our datasets. We did some experimental analyses of our datasets using different models (Gibbs vs. Metropolis-Hastings algorithms, Gamma vs. non-Gamma models) and compared the estimated $p$s, $Ts$s, and $Te$s. The results indicate that the Gibbs sampling can generate highly comparable $p$s, $Ts$s, and $Te$s to the Metropolis-Hastings algorithm[103] (Supplementary Fig. 29), especially when the Gamma model is not used (the estimates of these parameters are almost identical; Pearson's $r > 0.99990$, $p < 0.001$). As the Metropolis-Hastings algorithm can accurately recover the parameters of the preservation and birth-death processes[36,37], the preservation-corrected $Ts$s and $Te$s generated by the Gibbs-non-Gamma model and the origination and extinction rates subsequently estimated by LiteRate can also achieve this.

PyRate with the Gibbs algorithm was run for 10 million iterations. To reduce the size of log files generated, 1000 posterior samples were saved. All posterior samples were assessed using Tracer[106] (v.1.7.1). Convergence was confirmed if the effective sample size (ESS) was >200 for all parameters, and a proper burn-in percentage was suggested. For replications with ESSs <200, longer iterations were analysed. Mean times of origination and extinction were generated using the -ginput function in PyRate after a 10% burn-in, and then analysed by LiteRate. LiteRate was run for 200 million generations, saving 1000 posterior samples. Similarly, longer generations were run for replications that did not reach convergence. After a 10% burn-in, the posterior estimates across all replications were combined, then the mean and 95% highest posterior density intervals of origination, extinction, and net (i.e., origination minus extinction) rates were calculated (-combLog and -plotRJ functions in PyRate).

### Diversity estimation

We used two methods, the -ltt (lineage through time) function of PyRate and the recently developed Bayesian model-based mcmcDivE[60], to estimate the diversity of brachiopods and bivalves. The former is calculated based on $Ts$s and $Te$s generated by PyRate. The latter considers the uneven sampling of the fossil record and uses the preservation rates generated by PyRate and the occurrence data to estimate sampling-corrected diversity[60]. For all datasets (including global, ecological, and regional datasets), mcmcDivE was run for 2 million generations at 1 Myr intervals. Convergence was assessed using Tracer[106] (v.1.7.1). More generations were analysed if ESSs <200. The first 10% of the posterior samples were discarded as burn-in. The median and 95% highest posterior density intervals of diversity were calculated.

The two methods were developed based on completely different theory. Instead of considering the unsampled data between the first and last appearance of lineages (i.e., $Ts$s and $Te$s) in the -ltt function, the mcmcDivE was designed to compensate for taxa that are not sampled for a specific time period[60] (e.g., 1 Myr). We compared the results of these methods, finding that mcmcDivE tends to produce noisier diversity trajectories (Supplementary Fig. 30). According to simulated analyses, mcmcDivE was shown to be more accurate than other methods[60], but whether it still performs well with real data (with high temporal and spatial sampling bias) needs further confirmation. Because the following multivariate analyses use $Ts$s and $Te$s to decipher relationships between rates and factors, -ltt diversities were adopted in the MBD analyses.

### Potential drivers of diversification dynamics

Except for the possible brachiopod-bivalve interactions, other potential factors/predictors may also affect the evolutions of the two clades. Here we selected ten predictors (Supplementary Fig. 31), including biotic and abiotic factors to infer their influences on the diversification dynamics of brachiopods and bivalves.

Biotic factors include three categories, brachiopod and bivalve diversities, and predator diversity. The mean diversities generated by

the *-ltt* function were referred to as brachiopod and bivalve diversities. The intensity of predator-prey interactions is also an important factor that might affect the fates of brachiopods and bivalves[30,31,107]. Both brachiopods and bivalves are subject to predation[84,108–110], and the radiations of advanced predators, such as neogastropods, crabs, and teleost fishes were hypothesised to have caused the late-Mesozoic decline of brachiopods[30,31]. Previous studies showed that potential predators of the Permian–Jurassic shelled animals include echinoderms, arthropods, gastropods, fishes, and reptiles[13,84,108,110]. Here diversities of two benthic groups, arthropods and echinoderms, were selected to represent the diversity of predators. Fishes and reptiles were not considered because their diversity patterns are poorly known and their fossil records are prone to be strongly affected by sampling bias such as the Lagerstätte effect[111,112]. Besides, the largest group that consumes shelled animals today, the teleosts, were not diverse until the Late Cretaceous. In the class Gastropoda, although some clades are predatory, known carnivorous forms originated later than the Jurassic[110]. The possibly parasitic platyceratid gastropods occurred in the Palaeozoic, but they are very few in the Permian–Jurassic interval. Overall, only arthropods and echinoderms have a relatively continuous and abundant fossil record. Fossil occurrences of arthropods and echinoderms were downloaded from the PBDB on 23/11/2022. Similar to the post-Cambrian brachiopod and bivalve datasets, these data were also cleaned using the same procedure. For arthropods, the Eurypterida, Xiphosura, Decapoda, and Stomatopoda were regarded as potential predators and the data of other groups were discarded; for echinoderms, the occurrences of Asteroidea, Echinoidea, and Ophiuroidea were kept. The data of the two groups were analysed separately using PyRate (the method is identical to that for brachiopods and bivalves), and diversity was calculated using the *-ltt* function of PyRate. Then, the means of their estimated diversities were summed as a predictor to be used in multivariate analyses (Supplementary Fig. 31). It is difficult to avoid some biases in the estimate of predation intensity due to the uncertainty of trophic structure in ancient oceans and an incomplete fossil record. The curve we generated, although imperfect, nonetheless follows the expectation that predators became increasingly diverse in the Mesozoic.

The seven abiotic factors include seawater temperature, eustatic sea level, marine carbon isotope composition ($\delta^{13}C$), sulfur isotope composition ($\delta^{34}S$), strontium isotope composition ($^{87}Sr/^{86}Sr$), global continental fragmentation, and proportion of carbonate occurrences (Supplementary Fig. 31). These factors have been widely tested to explain the diversification dynamics of marine organisms including brachiopods and bivalves[20,65,67,113–117]. The trajectory of seawater temperature came from Scotese et al.[75]; their 'global average temperature' estimated from oxygen isotopes ($\delta^{18}O$) was adopted. The Permian eustatic sea level was from Haq and Schutter[118]. The Triassic and Jurassic sea level was from Haq[119,120]. The carbon isotope data were from Cramer and Jarvis[121], and these were smoothed using the locally-weighted polynomial regression[122]. The dataset of $\delta^{34}S$ was from Present et al.[123]. Isotopic values derived from bulk rock carbonate-associated sulfate were discarded because they show greater variability than isotopic values from other sources and may not reflect the $\delta^{34}S$ composition of palaeo-seawater sulfate[123]. Secular change curves of $^{87}Sr/^{86}Sr$ followed McArthur et al.[124]. The original global continental fragmentation index[117] was based on the Seton et al.'s[125,126] tectonic plate model. Here, we recalculated the index value based on the PALEOMAP tectonic model[127], to be consistent with that used in the regional analysis. A high value means all plates are not touching, and a low value suggests the presence of a supercontinent. The last factor, the proportion of carbonate occurrences, was selected to describe the habitat (substrate) condition. The vast majority of the post-Permian brachiopods were pedicle-attached. Mesozoic brachiopods became more and more frequent in carbonate lithologies which are on average firmer than siliciclastics, and the loss of habitats because of

strengthened bioturbation possibly drove the Mesozoic–Cenozoic decline of brachiopods[67]. To approximate habitat availability, we calculated the proportion of occurrences (of all brachiopods and bivalves) reported from carbonate lithologies at the substage level. This factor also indicates whether the temporal distributions of occurrences were affected by alternations of rock lithology[128], and these rate variations, therefore, might be attributed to the quality of fossil record. The proportional number of carbonate formations is not employed as the proxy because the number of occurrences varies greatly among formations. It is inappropriate to treat a highly fossiliferous formation and a fossil-poor unit as an equally weighted unit in the analyses, although the proportions of the formations correlate positively with occurrences of fossils (Spearman's $\rho = 0.39$, $p < 0.01$).

### Multivariate birth-death analyses

The multivariate birth-death (MBD) model was employed to assess the influence of biotic and abiotic factors on the diversification dynamics of brachiopods and bivalves in the software package PyRateMBD[38]. In this model, origination and extinction rates are correlated with time-continuous variables (i.e., factors, predictors) through an exponential function or linear function, so that the observed rates represent the function of the baseline rates, correlation parameters, and external factors. Using an MCMC algorithm, PyRateMBD can jointly estimate the baseline origination ($\lambda_O$) and extinction ($\mu_O$) rates, and correlation parameters ($G\lambda$ and $G\mu$). A positive $G$ implies that the factor correlates positively with the rate and vice versa. Besides, by using a horseshoe prior, the analysis can assess the support of the assumed relationship, preventing over-parameterisation. A correlation parameter with a high shrinkage weight ($w$) close to 0 is likely to be noise, while a value close to 1 represents a true signal[38].

We performed the multivariate analysis for five datasets. The first two are the whole Permian–Jurassic brachiopods and bivalves, and their diversities are included as predictors. This analysis is similar to previous studies[5,19,20,27] in treating bivalves as a whole group. The other three datasets analysed are epifaunal brachiopods (which shows very little difference from the whole brachiopod dataset because infaunal brachiopods are rather low in diversity), epifaunal bivalves, and infaunal bivalves, so the three diversity trajectories were included as predictors. Because of greater overlap in food resource and habitat, epifaunal bivalves were hypothesised as competitors of brachiopods[30,31]. However, no studies separately compared epifaunal and infaunal bivalve diversities with brachiopod diversities (although Sepkoski[19] noticed this problem). By dividing bivalves into two ecological groups, it will help to distinguish their potentially different relationship with brachiopods.

Before analysis, all factors were rescaled to 0 and 1 to remove biases from the absolute magnitudes of input values. By default, PyRate uses stepwise interpolation to connect the discrete values of the predictors. To precisely track the variation of the values, factors were instead linearly interpolated at 0.1 Myr intervals. We ran the MBD model for 30 million generations and saved 1000 posterior samples. For every dataset, ten replications were analysed separately (the *Ts*s and *Te*s estimated by PyRate were used as input, avoiding re-modelling the complex preservation process), and the ten log files generated were combined after a 10% burnin. Convergence was assessed using Tracer[106] (v.1.7.1), and longer generations were analysed for replications with ESSs <200. The 95% highest posterior density intervals of correlation parameters and mean shrinkage weights were calculated. If the shrinkage weight of a correlation is greater than 0.5, the relationship was regarded as significant[38].

As suggested by other authors[61,62], the drivers of diversification may change over time, and the relationships revealed over an extended period may fail in being significant within a short period. Therefore, we first conducted the MBD analyses for the whole Permian–Jurassic dataset. Next, the entire Permian–Jurassic interval

was subdivided into four shorter windows: Asselian–Wordian, Capitanian–Ladinian, Carnian–Toarcian, and Aalenian–Tithonian, and each was analysed using the MBD model. These time windows have different durations (from ~27.3 to ~62.3 Myr), but all are composed of six or seven stages/ages. Given that the age of occurrence in the datasets relies largely on stages/substages, subdivision of time bins based on the number of stages rather than the absolute range (e.g., 40 Myr) is more appropriate. For all MBD analyses, both exponential and linear models were run. We calculated Bayes factors to investigate the relative support of the two models following Lehtonen et al.[38]. Except for the Asselian–Wordian datasets which did not show clear support for either one model (absolute difference of the log Bayes factors[129] is <2), the exponential model was supported in all cases. Therefore, only the results of the exponential model were reported.

### Regional analyses

The global diversity and diversification dynamics represent the summed diversity across a set of geographically and environmentally distinct regions, and the real diversification process occurring in a region may be masked by the global trend[50,59,60]. To investigate whether diversification patterns of brachiopods and bivalves are different among geographical regions, we spatially standardised the data and analysed their respective rates. The workflow of spatial standardisation followed Flannery-Sutherland et al.[60]. Instead of dividing the regions arbitrarily, we made spatial windows according to the palaeobiogeographical pattern of fossil occurrences. First, the occurrences were spatially binned using a hexagonal grid through the *icosa* R package[130]. Second, a grid-taxon matrix was made based on the presence/absence of taxa in these grids. Last, we used two methods, network analysis and partitioning around medoids clustering[131], to identify biogeographic regions. Network analysis was implemented using the *igraph* R package[132]. Partitioning around medoids clustering was implemented using the *fpc* R package[133], and the modified Forbes index[134] was used to calculate the dissimilarity matrix for partitioning around medoids clustering. Then the result of grouping can be used as a reference to construct the spatial windows. This procedure was implemented in an R function, *compare_biogeography*, and can be directly applied to study the palaeobiogeography of other fossil datasets. Based on the analyses of brachiopod and bivalve occurrences (Supplementary Figs. 32–35), four biogeographical regions having relatively abundant and continuous fossil records (especially the Permian–Triassic interval) were selected to analyse the regional rates, including northern Panthalassa, north-western Tethys, south-western Tethys, and eastern Tethys (Supplementary Fig. 36).

Prior to spatial standardisation, the palaeo-coordinates of the fossil occurrences were updated based on the modern coordinates and revised midpoint ages, using the PALEOMAP tectonic model[127] through the GPlates Web Service (https://gwsdoc.gplates.org). A spatial window was constructed for each of the four regions using the *spacetimewind* function[60]. The generated spatial window is relatively stable in size, and moves over time to track the movement of the focal plate to guarantee that the analytical results were not biased by the species-area effect and continental drift. Fossil occurrence data were subsampled into each region and then standardised using the *spacetimestand* function[60]. We applied the minimum spanning tree (MST) length as the metric to standardise the occurrence data. In this procedure, fossil occurrences are binned into hexagonal grids, and an MST is reconstructed from the grid centres that contain fossil data. If the MST length is greater than the target value, the cells with the smallest amount of data are removed from the MST, until the threshold is reached[60]. We calculated the MST length of raw data in each time bin, and the median MST length of all time bins was adopted as the threshold. After spatial standardisation, the data show much less variance in MST length (Supplementary Fig. 37). For rate analysis, the PyRate + LiteRate procedure is inapplicable because many regions

have some bins that lack any records of fossil occurrences, and then LiteRate cannot estimate the rates successfully. As a result, only PyRate with the Gibbs algorithm was employed for the analyses. The rates estimated clearly show various patterns among regions despite the loss of some resolution.

### Diversification dynamics revealed by traditional methods

In addition to Bayesian model-based methods which estimate continuous rates, we also analysed the diversification dynamics of global brachiopods and bivalves using traditional methods (e.g., per-capita[41], three-timer[63], and its improved versions, gap-filler[135] and second-for-third[136]), which are based on discrete time bins. The workflow was identical to that of Kocsis et al.[40]. Every stage was regarded as a time bin, except the Neogene (four bins) and Quaternary (one bin). Occurrences were assigned to these time bins based on their ages, and those that could not be safely assigned to a certain bin were discarded. Rates and diversities were calculated using the *divDyn* R package[40]. Raw diversity and sampling-corrected diversity (using the Shareholder Quorum Subsampling algorithm[49]) were estimated by the two metrics, range-through and sampled-in-bin diversities (corrected by the three-timer sampling completeness[63]). Foote's per-capita rates[41] and Alroy's second-for-third rates[136] are reported (Supplementary Figs. 39–42). Compared with the per-capita rates of brachiopods and bivalves, the second-for-third rates are more volatile, especially for the post-Jurassic brachiopods. After checking the output of PyRate and *divDyn*, we attribute the high volatility of post-Jurassic brachiopod rates to the low preservation rate (sampling rate) during this time. Simulation studies have indicated that when diversity or sampling is low, three-timer and its related methods tend to generate results that are more volatile than the per-capita rate[37,135,137]. Generally, the per-capita rate is more accurate than other traditional methods[37,137], although it has its own drawbacks. Therefore, we did additional correlation tests on per-capita rates of brachiopods and bivalves. Rates in the first and last two bins were not included to remove the influence of edge-effects.

### Reporting summary

Further information on research design is available in the Nature Portfolio Reporting Summary linked to this article.

## Data availability

Most occurrence data were downloaded from the Paleobiology Database (https://paleobiodb.org/). In addition, we added some Permian–Jurassic fossil occurrences collected from primary references. All underlying data for this manuscript (including the URLs used to download the data, all newly added, raw and amended data analysed in this study) are available in Zenodo (https://doi.org/10.5281/zenodo.8216739)[138].

## Code availability

All R and Bash scripts used to revise the raw datasets and conduct our analyses are available in Zenodo (https://doi.org/10.5281/zenodo.8216739)[138].

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

## Acknowledgements

We are grateful to contributors of PBDB. We thank D. Silvestro for valuable discussions and assistance with PyRate. Y.P. Sun and S. Lee are thanked for providing important literature. The School of Earth Sciences, University of Bristol is acknowledged for hosting Z.G. as a visiting student for one year. This research was funded by NSFC grants (41930322, 41821001) to Z.-Q. C. and NERC grant (NE/IO27630/

1) and ERC Advanced Grant (788203 Innovation) to M.J.B. Z.G. was supported by China Scholarship Council (No. 202106410098) and Fundamental Research Funds for National Universities, China University of Geosciences (Wuhan). J.F.-S. was supported by NERC GW4+ DTP studentship (S100065-138/123).

## Author contributions

M.J.B. and Z.-Q.C. designed the research. Z.G. revised the data. Z.G. analysed the data with help from J.F.-S. J.F.-S. calculated the new continental fragmentation index. Z.G. drafted the paper with substantial input from all authors.

## Competing interests

The authors declare no competing interests.
