## [Peer Review File · Nature Communications]

Bayesian analyses indicate bivalves did not drive the downfall of brachiopods following the Permian-Triassic mass extinctionREVIEWER COMMENTS

Reviewer #1 (Remarks to the Author):

Review, Guo et al., MS# NCOMMS-23-12673-T

Summary: This is an ambitious paper, trying to tackle the recalcitrant problem of understanding the contrasting Phanerozoic diversity trajectories of bivalves and brachiopods. The paper sets the stage broadly, but while it makes clear and concise reference to the long (and diffuse) past literature on trying to understand the contrasting trajectories, it only incidentally (even if often) touches on these various previous ideas and data. Instead, it focuses on the authors' own assessment of the diversity dynamics (origination and extinction rates, and the resulting diversity trajectories) of the two groups, their correlations with each other and with a variety of environmental variables, touching on differences geographically, as well as breaking the analysis up ecologically. This a very ambitious research agenda. But in the end, this manuscript is too long, tackles too many issues, most of which are treated too diffusely, fails (for the most part) to make rich contact with the past literature, as well as having some methodological issues. They cover so much ground that I don't know what the non-specialist reader is supposed to take away from the paper that is new, and I think the discussion points are too simple for the specialist reader. Taken together, I feel these all combine to make the paper unsuitable for Nature Communications. Below I provide a little more detail on just some of these issues.

- 1) Abstract. The abstract reflects the diffuseness of the manuscript, viz (their text in italics): (a) Times of major biotic replacement have traditionally been interpreted in a Neodarwinian sense as mediated by innovation in the succeeding clades. I am not sure how useful this statement is, and is incomplete, for the traditional approach (is there any other?) also makes reference to changes in the environment (which they do point out in their text). (b) ... the switch from brachiopods to bivalves ... [has been] attributed to competitive exclusion of brachiopods by the superiorly adapted bivalves or simply to the fact that brachiopods had been hit especially hard by the PTME. OK, fine, but the idea of long-term competitive exclusion I do not think has any support. Then we change gears abruptly with (c) ... the two clades displayed highly comparable trends of diversification before the Jurassic (which I can't see in the data, and even if present, I don't understand the significance). Then back we go to the Permian-Triassic transition with (d) Insight from a multivariate birth-death model shows that the extinction of major brachiopod clades during the PTME set the stage for the BBS (I think we have known this for 50 years), with differential responses to high ocean temperatures post-extinction further facilitating their displacement by bivalves (ah, an interesting result, ignoring the fact that correlation is not causation). Then, a final conclusion, that largely confirms past conclusions (although contradicted by some of the literature which they don't resolve): (e) Our study indicates [I think 'confirms' would be a more accurate statement] that brachiopods and bivalves were not competitors over macroevolutionary timescales, with extinction events and environmental stresses shaping their divergent fates. This is so broad as to be not so informative. So, I don't see what non-specialists (or even specialists) are supposed to take away from this paper in its current form, although there are many aspects of analysis that will be of interest to specialists if presented more thoroughly in the specialist literature.
- 2) Similar trends between pre-Jurassic bivalve and brachiopod diversity dynamics. There are several issues here. First, this conclusion seems to be via visual inspection only, and I cannot

see it in the data, which makes we concerned that it is not true (unless the authors are simply referring to shared volatility compared with the post-Triassic records). This needs some quantification. Related to this, are they referring to just the largest changes, i.e., the fact that both groups respond to the mass extinctions? I would also note that even if the trajectories are in fact similar, this does not mean that the clades are not in competition (as implied in the abstract): there could have been biotic interaction between the two clades on shorter timescales, i.e., persistent competition on ecological timescales, with both responding similarly to larger perturbations (just as competitor companies might also respond similarly to global changes in consumer spending, thus showing similar large scale trajectories even though in competition). So, an analysis of the residuals might be of value here.

3) Methodological concerns:

- a) The brachiopod diversity trajectory (see Figure 1). This does not match my sense of their trajectory, largely due to the huge Permian peak. I suspect this is due to the authors addition of the very rich (largely Permian) Chinese data, which makes me suspect the curve is biased by heterogeneous temporal and spatial sampling.
- b) This then raises the general issue of how to deal with sampling biases. It appears that they have let PyrRate deal with preservation issues. It is crucial that the authors explore the robustness of their conclusions by also treating their data with some sort of sample standardization process, for example SQS, and then analyzing them with alternative approaches, for example divDyn, to compute origination and extinction rates, etc. That is, I simply don't know if I can trust the analysis – if more than one analytic approach leads to the same conclusions, that would greatly increase the power of the paper.

Reviewer #2 (Remarks to the Author):

Overall, the submitted manuscript is strong. The results are entirely predictable: the hypothesis that bivalves outcompeted brachiopods is theoretically reasonable, but has been clearly and unambiguously falsified since good global biodiversity data became available starting in the late 1970s. Nevertheless, there are practitioners who continue to promote (without evidence) this outdated hypothesis. In this context, the manuscript is important. Even though, the results are expected, presenting new lines of evidence that bivalves did not outcompete brachiopods after the Permian mass extinction (and vice versa before the extinction) is a valuable contribution to the literature.

The methods are strong and the writing is clear. I have a few minor suggestions, but I don't have any major reservations or concerns with the manuscript.

The most serious comment I have is that I think the language regarding the patterns of bivalve and brachiopod diversification before and after the Jurassic is greatly overstated. The first paragraph of the "Diversification dynamics of ..." Section on page 5 states that both clades show similar dynamics before the Jurassic, but the trends become distinct after the Jurassic. I don't think this claim is supported by Figure 2. My read on figure 2 is that the the Jurassic is, indeed, an inflection point in the macroevolutionary history of the two clades. However, the dynamics flip around that point. Before the Jurassic, the bivalves show relatively stable rates (Fig 2a and 2b) while the Brachiopod rates are volatile (Fig 2d and 2e). After the Jurassic, they flip and the

bivalves are volatile and the brachiopods are stable. I don't think this negates the overall interpretation of the data, but it does require a bit more explanation. I wonder if it has to do with the overall diversity? Does the birth-death model employed inherently produce more volatile rates when diversity is greater than 200-300 (genuine question)? Alternatively, perhaps asking reader to visually inspect raw time series is not the best way to evaluate the relative similarity of the trends. Might a cross plot of bivalve vs. brachiopod rates would be more effective??

Minor Comments & Suggestions:

I don't like the first part of the title "Red in tooth and gill". This is a variation on a common phrase, but is culturally specific and won't make sense to all readers. Admittedly, though I am familiar with the original phrase "red in tooth and claw", I had to look up the exact meaning. Even if this stems primarily from ignorance on my part, I suspect it will be unfamiliar to a large portion of Nature Communication's readership who do not speak English as their native language.

One of the things that plagues paleontology is the overuse of acronyms. I think the manuscript uses a few too many acronyms. Specifically, I feel the use of BBS for brachiopod-bivalve switch is gratuitous: I recommend dropping the acronym and just writing out brachiopod-bivalve switch (or transition). Moreover, the methods have turned into a bit of an acronym bonanza. The use of some is fine, but I would, again, recommend spelling out some of the acronyms that are only used a couple of times (e.g., ESS, HDPI, PAM). There are just a lot of acronyms to remember and it's a distraction while reading to have to stop and either think about what they mean or go back in the text to find their first use.

There are a number of places in the manuscript where two items are listed and then the second item is referred to as "latter" in a subsequent clause. As with the acronyms, this can be a distraction to the reader, especially if "latter" does not occur directly after the list. I recommend writing out the object being referred to where possible. A good example is on Page 8 Lines 190-192. "...Cretaceous and Cenozoic, which the latter ...". I was initially confused because I thought latter referred to the Cenozoic. I had to reread the sentence to remind myself that it was brachiopods being referenced.

Page 5 Line 54: I think "largest events in the history of marine life" should reference the end-Permian mass extinction, not the "BBS".

All time series figures: Please drop the vertical gray bars marking alternating time intervals. These are chart junk and make the plots difficult to read without conveying any important information. Important events are directly labeled (e.g., PTME), which render the gray bars superfluous. In addition, the yellow lines and labels for the PTME, etc. would be easier to read if they were a bit darker.

Page 7 Line 170: I think a word other than "observations" should be used here. I don't think the observations made in this analysis strengthen or weaken previous observations. Previous observations have already been made and are forever fixed. It's the interpretations of those observations that are strengthened, weakened, supported, etc.

Page 9 Line 218: I don't think "greater" is the best word to use here. I think "more" or "a greater number" would be better. Greater implies better or larger in size.

Page 15 First Paragraph. I think a very short discussion, or at least a citation, of the very important paper by Adrain and Westrop (2000, *Science: An empirical assessment of toxic paleobiology*) is appropriate here.

Page 17 Line 429. I humbly request that the authors add the > 6000 fossil occurrences mentioned here to the PBDB. This is, of course, not a requirement for publication and should not be considered in the editorial decision, but it would be a great service to the community. If the authors contact the database executive committee they can help arrange a bulk upload from excel/csv files so that the data does not need to be keystroked into the database by hand.

Page 21 Line 544: "PyRate considers the diversification of clades". What are the clades here? Are they simply Bivalvia vs. Brachiopoda or are rates being calculated for orders, families, etc. I think clarification on this point would be helpful.

Page 22 Line 587: Should "modal" be "model"?

Page 22 Line 591: "Fake" is not the right word to use here. Fake implies something that was created intentionally as a deception. I think spurious is a better word choice: it has a standard statistical usage that makes sense in this context.

Page 25 Line 663: I am a little confused why Echinoidea and Ophiuroidea were included as predators of bivalves and brachiopods. While it's likely that both may consume shelled benthic invertebrates (particularly larval/juvenile stages), I don't think they constitute a major source of food for either group: I am not convinced they should be included. If there is evidence/literature contrary to my understanding, please include a justification for their inclusion.

-Noel Heim

RESPONSE TO REVIEWER COMMENTS

Reviewer #1 (Remarks to the Author):

Summary: This is an ambitious paper, trying to tackle the recalcitrant problem of understanding the contrasting Phanerozoic diversity trajectories of bivalves and brachiopods. The paper sets the stage broadly, but while it makes clear and concise reference to the long (and diffuse) past literature on trying to understand the contrasting trajectories, it only incidentally (even if often) touches on these various previous ideas and data. Instead, it focuses on the authors' own assessment of the diversity dynamics (origination and extinction rates, and the resulting diversity trajectories) of the two groups, their correlations with each other and with a variety of environmental variables, touching on differences geographically, as well as breaking the analysis up ecologically. This a very ambitious research agenda. But in the end, this manuscript is too long, tackles too many issues, most of which are treated too diffusely, fails (for the most part) to make rich contact with the past literature, as well as having some methodological issues. They cover so much ground that I don't know what the non-specialist reader is supposed to take away from the paper that is new, and I think the discussion points are too simple for the specialist reader. Taken together, I feel these all combine to make the paper unsuitable for Nature Communications.

REPLY: Many thanks for your valuable comments on our study. We note that you are concerned about: 1) lack of focus (diffuseness), 2) too many points, and 3) omission of some important references to the earlier literature. We have considered all of these points carefully and revised the manuscript thoroughly.

We added some discussions about our new results and previous hypotheses, especially these three important BBS scenarios: Sepkoski (1996; competition hypothesis), Steele-Petrović (1979; physiological superiority of bivalves), and Liow et al. (2015; the latest time-series analyses showing the competition between the two). Additional important references have been cited in the revised version.

We re-analysed the same dataset using traditional methods (see responses below). The per-capita rates largely agree with the rates derived from the PyRate analysis. Moreover, correlation tests of these rates support our conclusion that the pre-Jurassic rates of brachiopods and bivalves have similar trends.

We also performed additional MBD analyses on the three groups: epifaunal brachiopods, epifaunal bivalves, and infaunal bivalves. Because of larger overlap in habitat and feeding mechanism, epifaunal bivalves were hypothesized to be competitors of brachiopods. The independent multivariate analysis of epifaunal and infaunal bivalves is an interesting issue that was overlooked by previous studies. These analyses also connect the 'Rates of different ecological groups' section and the multivariate section, making the manuscript more coherent.

Below I provide a little more detail on just some of these issues.

1) Abstract. The abstract reflects the diffuseness of the manuscript, viz (their text in italics):
(a) Times of major biotic replacement have traditionally been interpreted in a Neodarwinian sense as mediated by innovation in the succeeding clades. I am not sure how useful this

statement is, and is incomplete, for the traditional approach (is there any other?) also makes reference to changes in the environment (which they do point out in their text).

REPLY: Thanks, we have revised the statements.

(b) ... the switch from brachiopods to bivalves ... [has been] attributed to competitive exclusion of brachiopods by the superiorly adapted bivalves or simply to the fact that brachiopods had been hit especially hard by the PTME. OK, fine, but the idea of long-term competitive exclusion I do not think has any support.

REPLY: Although the brachiopod-bivalve switch is a classic example of biotic replacement, only a few studies have investigated their long-term diversification dynamics in detail. Gould and Calloway (1980) denied the presence of long-term competition between the two groups. In contrast, Sepkoski (1996), Liow et al. (2015), and Reitan and Liow (2017) re-studied this issue later on, and these authors found some evidence, and clarified the idea of competitive replacement between brachiopods and bivalves. Whether or not the evidence is reliable, the answer to this issue still remains open to debate. We agree with you that the idea of long-term competitive exclusion has been rarely supported, but obviously, new studies based on the updated datasets using advanced analytical methods are needed.

Then we change gears abruptly with (c) ... the two clades displayed highly comparable trends of diversification before the Jurassic (which I can't see in the data, and even if present, I don't understand the significance).

REPLY: Your suggestions are followed in the revised version (See responses below).

Then back we go to the Permian-Triassic transition with (d) Insight from a multivariate birth-death model shows that the extinction of major brachiopod clades during the PTME set the stage for the BBS (I think we have known this for 50 years), with differential responses to high ocean temperatures post-extinction further facilitating their displacement by bivalves (ah, an interesting result, ignoring the fact that correlation is not causation).

REPLY: Thanks, we agree with you that correlation is not causation. We have re-worded some sentences in the manuscript.

Then, a final conclusion, that largely confirms past conclusions (although contradicted by some of the literature which they don't resolve): (e) Our study indicates [I think 'confirms' would be a more accurate statement] that brachiopods and bivalves were not competitors over macroevolutionary timescales, with extinction events and environmental stresses shaping their divergent fates. This is so broad as to be not so informative. So, I don't see what non-specialists (or even specialists) are supposed to take away from this paper in its current form, although there are many aspects of analysis that will be of interest to specialists if presented more thoroughly in the specialist literature.

REPLY: It is true that our results 'confirm' the previous hypothesis rather than proposing a new scenario on the brachiopod-bivalve replacement. However, the previous studies (Gould and Calloway, 1980; Sepkoski, 1996) usually utilize diversity data, rather than diversification dynamics to figure out this issue. The same diversity trajectory can be achieved by completely different combinations of origination and extinction rates. Thus, disentangling the

two components is important to answer this question. Liow et al. (2015) did so, but they concluded that both brachiopods and bivalves are competitors, denying the conclusion given by Gould and Calloway (1980). In addition to competition, other factors have also been proposed to explain brachiopod and bivalve extinction or origination, especially across the PTME. However, most of these hypotheses were not tested in a multivariate analysis. In recent years, more accurate and advanced methods such as PyRate have been developed. We employ these new methods to re-study the long-debated brachiopod-bivalve switch issue.

Therefore, although our analyses did not lead to startling or shocking new conclusion, we argue for the merits of our work in two ways: (a) the brachiopod vs. bivalve replacement is a textbook example, widely known to many evolutionary biologists as well as palaeobiologists, and so our work will be important to those general readers; and (b) we reach a decisive conclusion based on use of the most detailed data set ever deployed and the widest array of state-of-the-art techniques, resolving a long-term but unresolved debate point. So, we have a headline that will startle many analysts and commentators in evolutionary study: brachiopods were not driven to extinction by ‘superior’ bivalves.

2) Similar trends between pre-Jurassic bivalve and brachiopod diversity dynamics. There are several issues here. First, this conclusion seems to be via visual inspection only, and I cannot see it in the data, which makes me concerned that it is not true (unless the authors are simply referring to shared volatility compared with the post-Triassic records). This needs some quantification. Related to this, are they referring to just the largest changes, i.e., the fact that both groups respond to the mass extinctions? I would also note that even if the trajectories are in fact similar, this does not mean that the clades are not in competition (as implied in the abstract): there could have been biotic interaction between the two clades on shorter timescales, i.e., persistent competition on ecological timescales, with both responding similarly to larger perturbations (just as competitor companies might also respond similarly to global changes in consumer spending, thus showing similar large scale trajectories even though in competition). So, an analysis of the residuals might be of value here.

REPLY: Thank you for your comments and suggestions. It is true that the “similar trend” that we stated before was based on visual inspection. Unlike traditional methods that estimate rates for every time bin, PyRate with the rjMCMC algorithm estimates the number and temporal placement of statistically significant rate changes, which reduces the risk of over-parameterisation. Therefore, the rates can be “flat” if there is no significant rate shift, and the abundant tied data renders direct and effective correlation analysis.

Following your suggestions below, we calculated diversification rates using traditional methods, and we also performed additional analyses to the per-capita rates. Correlation tests of these rates agree with our previous conclusion that origination and extinction rates of brachiopods and bivalves correlate positively with one another, and this relationship is more prominently demonstrated by the pre-Jurassic data.

We agree with you that there could be biotic interactions between the two clades in some cases, as Thayer’s experiments showed. However, whether these sporadic, localized events were the main cause of the macroevolutionary decline of brachiopods is questionable. Our study focuses on large-scale diversification dynamics.

Your suggestions on the analysis of residuals are also highly appreciated. Following your advice, we performed the reduced major axis regression analysis on their extinction and origination rates, respectively, and then calculated the residuals (Fig. S49). Clearly, the residuals do not correlate with brachiopod or bivalve diversities ($p > 0.7$), i.e., an unnoticeable role of competition on macroevolutionary timescales.

3) Methodological concerns:

a) The brachiopod diversity trajectory (see Figure 1). This does not match my sense of their trajectory, largely due to the huge Permian peak. I suspect this is due to the authors addition of the very rich (largely Permian) Chinese data, which makes me suspect the curve is biased by heterogeneous temporal and spatial sampling.

REPLY: Many thanks for your comments. Overall, the diversity trajectory of brachiopods shown in Figures 1 and 2 is close to the trajectories that were previously published elsewhere, including those based on:

- (1) PBDB database and “sample in bin” diversity (e.g., Alroy, 2010, sampling standardised using the SQS algorithm; Close et al., 2020, spatially and SQS standardised);
- (2) PBDB database and “range through” diversity (e.g., Payne et al., 2014);
- (3) well-curated databases and “range through” diversity (e.g., Sepkoski’s Compendium [Sepkoski, 1996]; Treatise on Invertebrate Paleontology [Curry and Brunton, 2007]).

All these trajectories show high peaks of diversity in the Ordovician–Devonian and Permian (see the figure below), although the peaks have various heights.

Diversity peaks of Permian brachiopods published in previous literature (Sepkoski, 1996; Curry and Brunton, 2007; Alroy, 2010; Payne et al., 2014; Close et al., 2020). Diversity curves of Alroy (2010) and Close et al. (2020) are sampling standardised. Colours in the diagram of Close et al. (2020) represent diversity in different paleolatitude zones.

The fossil occurrences that we added indeed do not increase the steepness of the diversity peak of Permian brachiopods, because the curves in Figures 1 and 2 were totally based on the raw data downloaded from the PBDB. These curves are used to reflect the long-term diversification dynamics of brachiopods and bivalves. For the subsequent Permian-Jurassic analysis, we did many revisions including adding additional occurrences. Despite all this, we found that only two of the Permian genera that we added are absent in PBDB.

We compared the taxonomic compositions in the raw PBDB dataset, the revised PBDB dataset, and those in the Treatise, and found that the high Permian peak of brachiopod diversity in Figures 1 and 2 is mainly caused by two types of taxa. First, the raw PBDB dataset contains 89 Permian genera that should be discarded because these taxa are invalid, or synonymous to other taxa, or should not occur in the Permian (i.e., identifications of the fossils are problematic). Second, many new Permian genera have been established or re-validated in recent years after the Treatise of Brachiopoda was published in 2000s. We counted the number of genera that are recorded in the PBDB but were not included in the Treatise of Brachiopoda, and found that 114 Permian genera were established or re-validated after the Treatise was published. In contrast, there are 31 and 22 new genera in Devonian and Carboniferous, respectively that were not included in the Treatise. As a result, given that so many new genera were established from the Permian fossil records, the Permian peak of diversity should be higher than the diversity peak of the same age shown by Curry and Brunton (2007; see figure above). It is noteworthy that when the diversity is calculated for SQS subsampled data (Fig. S39), the Permian diversity peak is still prominent.

In summary, the pronounced peak of Permian diversity is a character of the PBDB dataset itself, and does not appear to be a bias that arose from the various analytical methods that we employed. When the raw dataset is taxonomically emended, the Permian peak is less steep (the highest Permian diversity is <400 in Figure 3). This highlights the importance of taxonomic emendation for the datasets that are used for big data analysis, as we did for the Permian-Jurassic dataset. It should be noted that although the raw post-Cambrian diversity is less accurate, it does not influence our subsequent analyses. The multivariate analysis is based on the Permian-Jurassic diversities which were calculated from the taxonomically revised datasets.

Refs cited in this Reply:

- Alroy J. The shifting balance of diversity among major marine animal groups. *Science* **329**, 1191–1194 (2010).
- Carlson SJ. The evolution of Brachiopoda. *Annual Review of Earth and Planetary Sciences* **44**, 409–438 (2016).
- Close RA, Benson RBJ, Saupe EE, Clapham ME, Butler RJ. The spatial structure of Phanerozoic marine animal diversity. *Science* **368**, 420–424 (2020).
- Curry GB, Brunton CHC. Stratigraphic distribution of brachiopods. In: *Treatise on Invertebrate Paleontology, Part H (Revised), Brachiopoda, Vol. 6.* (ed Selden PA). Geological Society of America and University of Kansas (2007).
- Payne JL, Heim NA, Knope ML, McClain CR. Metabolic dominance of bivalves predates brachiopod diversity decline by more than 150 million years. *Proceedings of the Royal Society B: Biological Sciences* **281**, 20133122 (2014).

Sepkoski Jr, JJ. Competition in macroevolution: the double wedge revisited. In: *Evolutionary Paleobiology* (eds Jablonski D, Erwin DH, Lipps JH). University of Chicago Press (1996).

b) This then raises the general issue of how to deal with sampling biases. It appears that they have let PyRate deal with preservation issues. It is crucial that the authors explore the robustness of their conclusions by also treating their data with some sort of sample standardization process, for example SQS, and then analyzing them with alternative approaches, for example divDyn, to compute origination and extinction rates, etc. That is, I simply don't know if I can trust the analysis – if more than one analytic approach leads to the same conclusions, that would greatly increase the power of the paper.

REPLY: It is true that incompleteness of fossil records and sampling biases are often an important problem and tough issue for fossil record-based statistical analysis and modelling. Indeed, the great performance of PyRate has been tested repeatedly (although some papers questioned the accuracy of PyRate; see response and discussion in Flannery-Sutherland et al. [2022]). Analyses of simulated datasets indicated that PyRate can unravel the underlying diversification processes (Silvestro et al., 2014) under a variety of preservation scenarios. Even if a large fraction of taxa is not sampled, PyRate is also able to reveal some true signals, and the confidence intervals cover the real trajectory.

PyRate has many advantages over traditional methods (Silvestro et al., 2014, 2019), and its Bayesian framework allows for testing the significance of rate shifts. Silvestro et al. (2019) performed many simulation tests, and showed that the relative error of PyRate results is much lower than those produced by traditional methods (e.g., Alroy's three-timer, Foote's boundary-crossing or per-capita). Traditional methods based on discrete time slices usually generate more volatile curves when the true evolutionary rates are constant (Silvestro et al., 2019), reflecting that traditional methods may have the problem of over-parameterisation. Although these uncertainties can be represented by confidence intervals from multiple analyses, their widths are usually greater than those of PyRate, especially when diversity is low.

Recently, Warnock et al. (2020) proposed a fossilised birth-death (FBD) model to estimate the diversification of fossil data. Their analysis of simulated data suggested that the FBD model is more accurate than the birth-death (BD) model that PyRate uses. However, in most cases, both BD and FBD models generate comparable results, which are better than that derived from traditional methods (especially the three-timer) (Warnock et al., 2020). The FBD model has been presented in PyRate, but, due to its complexity, this model is difficult to be applied to our large datasets. Therefore, we believe that PyRate with the BD model is currently the most suitable method for the analysis of our datasets.

Following your suggestion, we have re-analysed the diversification rates using the traditional methods (those provided in the divDyn package) and diversity. It is expected that different methods will generate slightly different results. Here we report the per-capita and second-for-third (the improved version of three-timer and gap-filler) rates. An obvious difference between the two is, the second-for-third rate of brachiopods is highly volatile after the Jurassic, but the per-capita rate in the same period is low, like our PyRate result. We checked the divDyn and PyRate output files and attribute this difference to the low

preservation rate of brachiopods at that time (the low sampling rate of Cenozoic brachiopods was also reported by Liow et al. (2015)). Analyses of simulated data have indicated that when diversity or sampling rate (preservation rate) is low, three-timer or its related methods tend to generate results that are more volatile and less accurate than the per-capita method (Alroy, 2014; Silvestro et al., 2019; Warnock et al., 2020). Although the per-capita rate (and every other method) has its own drawbacks, we used it to make additional comparisons and tests (see Supplementary Figures). Generally, the per-capita rates estimated here show largely comparable trends with our PyRate trajectories.

Refs cited in this Reply:

Alroy, J. Accurate and precise estimates of origination and extinction rates. *Paleobiology* 40, 374–397 (2014).

Flannery-Sutherland JT, Silvestro D, Benton MJ. Global diversity dynamics in the fossil record are regionally heterogeneous. *Nature Communications* 13, 2751 (2022).

Silvestro D, Schnitzler J, Liow LH, Antonelli A, Salamin N. Bayesian estimation of speciation and extinction from incomplete fossil occurrence data. *Systematic Biology* 63, 349–367 (2014).

Silvestro D, Salamin N, Antonelli A, Meyer X. Improved estimation of macroevolutionary rates from fossil data using a Bayesian framework. *Paleobiology* 45, 546–570 (2019).

Warnock RCM, Heath TA, Stadler T. Assessing the impact of incomplete species sampling on estimates of speciation and extinction rates. *Paleobiology* 46, 137–157 (2020).

Reviewer #2 (Remarks to the Author):

Overall, the submitted manuscript is strong. The results are entirely predictable: the hypothesis that bivalves outcompeted brachiopods is theoretically reasonable, but has been clearly and unambiguously falsified since good global biodiversity data became available starting in the late 1970s. Nevertheless, there are practitioners who continue to promote (without evidence) this outdated hypothesis. In this context, the manuscript is important. Even though, the results are expected, presenting new lines of evidence that bivalves did not outcompete brachiopods after the Permian mass extinction (and vice versa before the extinction) is a valuable contribution to the literature.

The methods are strong and the writing is clear. I have a few minor suggestions, but I don't have any major reservations or concerns with the manuscript.

REPLY: Many thanks for the positive comments.

The most serious comment I have is that I think the language regarding the patterns of bivalve and brachiopod diversification before and after the Jurassic is greatly overstated. The first paragraph of the "Diversification dynamics of ..." Section on page 5 states that both clades show similar dynamics before the Jurassic, but the trends become distinct after the Jurassic. I don't think this claim is supported by Figure 2. My read on figure 2 is that the the Jurassic is, indeed, an inflection point in the macroevolutionary history of the two clades. However, the dynamics flip around that point. Before the Jurassic, the bivalves show

relatively stable rates (Fig 2a and 2b) while the Brachiopod rates are volatile (Fig 2d and 2e). After the Jurassic, they flip and the bivalves are volatile and the brachiopods are stable. I don't think this negates the overall interpretation of the data, but it does require a bit more explanation. I wonder if it has to do with the overall diversity? Does the birth-death model employed inherently produce more volatile rates when diversity is greater than 200-300 (genuine question)? Alternatively, perhaps asking reader to visually inspect raw time series is not the best way to evaluate the relative similarity of the trends. Might a cross plot of bivalve vs. brachiopod rates would be more effective??

REPLY: Thanks for your comments. We are sorry that we did not do correlation test of these rates. Unlike traditional methods that estimate rates for every time bin, PyRate with the rjMCMC algorithm estimates the number and temporal placement of statistically significant rate changes, which reduces the risk of over-parameterisation. Therefore, the rates can be “flat” if there is no significant rate shift, and the abundant tied data renders direct and effective correlation analysis.

Silvestro et al. (2014, 2019) indicated that PyRate can extract the underlying diversification dynamics even if the preservation rate or diversity is low. When the preservation rate is low, although it cannot accurately estimate the absolute value of rates, it can still detect the time of rate shifts. In such a case, the final mean rates may be flatter, but the wide confidence interval still covers the real rates. Accordingly, the flat brachiopod rates with narrow confidence intervals in the Cenozoic should be a true signal. The low diversity is also not a problem. Many previous studies using PyRate focused on clades with low diversities and rate shifts were successfully recovered. Some fossil groups that we studied (e.g., cemented bivalves and deep infaunal bivalves shown in Fig. 4) also have very low diversity, but the mean rates have many fluctuations.

This is also the reason why we said the trends of brachiopod and bivalve diversification rates are “similar” to one another. The bivalve rate is much less volatile than the brachiopod rate in the early–middle Paleozoic, but at the same time, the confidence interval is wide. The low volatility of the bivalve rate in this interval is possibly caused by the low bivalve preservation rate, but the long-term general trend is similar to brachiopods: both clades had rates gradually decreasing in Ordovician–Devonian.

We also calculated their diversification rates using traditional methods following Reviewer 1’s suggestion (see responses above), and the result allows for a correlation test. It is shown that before the Jurassic, brachiopod and bivalve rates are significantly correlated, but the Jurassic–Quaternary rates are less correlated. This provides additional evidence for our previous conclusions.

Refs cited in this Reply:

Silvestro D, Schnitzler J, Liow LH, Antonelli A, Salamin N. Bayesian estimation of speciation and extinction from incomplete fossil occurrence data. *Systematic Biology* **63**, 349–367 (2014).

Silvestro D, Salamin N, Antonelli A, Meyer X. Improved estimation of macroevolutionary rates from fossil data using a Bayesian framework. *Paleobiology* **45**, 546–570 (2019).

Minor Comments & Suggestions:

I don't like the first part of the title "Red in tooth and gill". This is a variation on a common phrase, but is culturally specific and won't make sense to all readers. Admittedly, though I am familiar with the original phrase "red in tooth and claw", I had to look up the exact meaning. Even if this stems primarily from ignorance on my part, I suspect it will be unfamiliar to a large portion of Nature Communication's readership who do not speak English as their native language.

REPLY: Thanks, we have removed this part from the title.

One of the things that plagues paleontology is the overuse of acronyms. I think the manuscript uses a few too many acronyms. Specifically, I feel the use of BBS for brachiopod-bivalve switch is gratuitous: I recommend dropping the acronym and just writing out brachiopod-bivalve switch (or transition). Moreover, the methods have turned into a bit of an acronym bonanza. The use of some is fine, but I would, again, recommend spelling out some of the acronyms that are only used a couple of times (e.g., ESS, HDPI, PAM). There are just a lot of acronyms to remember and it's a distraction while reading to have to stop and either think about what they mean or go back in the text to find their first use.

REPLY: Many thanks. We have deleted the unnecessary acronyms and spelled out some of the acronyms.

There are a number of places in the manuscript where two items are listed and then the second item is referred to as "latter" in a subsequent clause. As with the acronyms, this can be a distraction to the reader, especially if "latter" does not occur directly after the list. I recommend writing out the object being referred to where possible. A good example is on Page 8 Lines 190-192. "...Cretaceous and Cenozoic, which the latter ...". I was initially confused because I thought latter referred to the Cenozoic. I had to reread the sentence to remind myself that it was brachiopods being referenced.

REPLY: Thanks, we have revised them.

Page 5 Line 54: I think "largest events in the history of marine life" should reference the end-Permian mass extinction, not the "BBS".

REPLY: Thanks for the comments. We have revised it.

All time series figures: Please drop the vertical gray bars marking alternating time intervals. These are chart junk and make the plots difficult to read without conveying any important information. Important events are directly labeled (e.g., PTME), which render the gray bars superfluous. In addition, the yellow lines and labels for the PTME, etc. would be easier to read if they were a bit darker.

REPLY: Thanks very much for your suggestions. We have removed all the gray bars in the main figures to make them clearer. But we have kept them in the Supplementary Figures to make it easy for the reader to compare these time series.

Page 7 Line 170: I think a word other than "observations" should be used here. I don't think the observations made in this analysis strengthen or weaken previous observations. Previous

observations have already been made and are forever fixed. It's the interpretations of those observations that are strengthened, weakened, supported, etc.

REPLY: Thanks for your comments. We have deleted this statement.

Page 9 Line 218: I don't think "greater" is the best word to use here. I think "more" or "a greater number" would be better. Greater implies better or larger in size.

REPLY: Thanks very much. We have corrected it.

Page 15 First Paragraph. I think a very short discussion, or at least a citation, of the very important paper by Adrain and Westrop (2000, Science: An empirical assessment of toxic paleobiology) is appropriate here.

REPLY: Thanks for your suggestions. We have cited this paper.

Page 17 Line 429. I humbly request that the authors add the > 6000 fossil occurrences mentioned here to the PBDB. This is, of course, not a requirement for publication and should not be considered in the editorial decision, but it would be a great service to the community. If the authors contact the database executive committee they can help arrange a bulk upload from excel/csv files so that the data does not need to be keystroked into the database by hand.

REPLY: Thanks for your suggestions. The additional occurrence data that we added have been provided in Supplementary Material. They have identical column names with PBDB data and can be easily read and manipulated in R. We will try to upload these occurrences to PBDB so it will be good for future studies.

Page 21 Line 544: "PyRate considers the diversification of clades". What are the clades here? Are they simply Bivalvia vs. Brachiopoda or are rates being calculated for orders, families, etc. I think clarification on this point would be helpful.

REPLY: Apologies for this confusion. It should be "a group". The diversification of any groups is regarded as a continuous process by PyRate.

Page 22 Line 587: Should "modal" be "model"?

REPLY: Thanks. Corrected.

Page 22 Line 591: "Fake" is not the right word to use here. Fake implies something that was created intentionally as a deception. I think spurious is a better word choice: it has a standard statistical usage that makes sense in this context.

REPLY: Thanks. Corrected.

Page 25 Line 663: I am a little confused why Echinoidea and Ophiuroidea were included as predators of bivalves and brachiopods. While it's likely that both may consume shelled benthic invertebrates (particularly larval/juvenile stages), I don't think they constitute a major source of food for either group: I am not convinced they should be included. If there is evidence/literature contrary to my understanding, please include a justification for their inclusion.

REPLY: Thanks for your comments. We agree that bivalves and brachiopods both are probably not major foods of the Echinoidea and Ophiuroidea. But it is clear that Echinoidea can consume bivalves when they lived together (Penchaszadeh et al., 2004). Bivalves or brachiopods may not be a big component of their food resource, but Echinoidea can also increase brachiopod mortality by grazing (Tomašových, 2008), which is also an important mechanism of the Mesozoic marine revolution (Vermeij, 1977). For the Ophiuroidea, it seems that both brachiopods and bivalves are not major sources of food. However, their predation on brachiopods is repeatedly mentioned in papers and the Treatise (Fouke and LaBarbera, 1986), and therefore, we included them in the analyses. The Ophiuroidea is only a small part of the predators. Its diversity doesn't show rapid changes. The rise of the predator diversity trajectory in the Middle–Late Jurassic is largely contributed by crustaceans and echinoids. In fact, it is very difficult to estimate the predation intensity in ancient ecosystems, and the proportions of contributions from different groups are difficult to estimate, so we only roughly estimate the predation pressure by adding diversities of multiple predatory groups. We admit that this curve is far from perfect, but we think that it should be more or less useful in a long-window analysis.

Fouke, B.W., and LaBarbera, M. Ecology of shallow water brachiopods in Jamaica: Proof of predation and its implications. 4th North American Paleontological Convention Abstracts 4, A16 (1986).

Penchaszadeh, P. E., Bigatti, G. & Miloslavich, P. Feeding of *Pseudechinus magellanicus* (Philippi, 1857) (Echinoidea: Temnopleuridae) in the SW Atlantic coast (Argentina). *Ophelia* 58, 91–99 (2004).

Tomašových, A. Substrate exploitation and resistance to biotic disturbance in the brachiopod *Terebratalia transversa* and the bivalve *Pododesmus macrochisma*. *Marine Ecology Progress Series* 363, 157–170 (2008).

Vermeij, G. J. The Mesozoic marine revolution: Evidence from snails, predators and grazers. *Paleobiology* 3, 245–258 (1977).

-Noel Heim

REVIEWERS' COMMENTS

Reviewer #1 (Remarks to the Author):

Summary: This is the second time I have seen this paper. I felt the first version was too diffuse, did not make proper contact with the literature, and coupled with my feeling that this issue was already settled, to warrant publication in this venue. However, upon reading this revision, I am convinced that this is indeed should be published in Nature Communications – it is a very good paper, with several sophisticated yet clearly expressed arguments relevant to establishing causality of diversity change in the fossil record, not just for bivalves versus brachiopods, but in general. This is an important paper, well executed.

Below are a series of largely minor comments that I think will help make the paper more readable (frankly, I don't know how anyone can write a sophisticated paper in another language, but these authors have done a fine job). In the interests of full disclosure, I did not scrutinize the methods section.

Suggestions for improvement

Lines 21-22: The authors' say that "we find that unexpectedly the two clades displayed highly comparable trends of diversification before the Jurassic". It seems to me that the details are actually quite different (I don't see the 'highly comparable trends'), but they DO share similar same large-scale trends, so perhaps this generality ('similar large-scale trends') is the phrase that belongs in the abstract (not 'highly comparable trends').

Line 38: The reference to the 'Big Five' (Raup and Sepkoski 1982) is over 40 years old; maybe the authors' could cite the 2023 review of the Big Five in the new journal "Cambridge Prisms: Extinction".

Line 79: 'dependence' mis-spelled.

Line 92: Replace 'any difference arising between' with 'differences between'.

Line 102: I still can't see the 'comparable trends of diversification dynamics, especially before the Jurassic' in the Figure (my eye is going to the fine detail of the wiggles). But if the next sentence explains what you mean, then I agree. So maybe just insert 'viz.' between the sentences to indicate that you are going to tell is in what way they are similar – then I am OK with the statement.

Line 116: Insert 'their' before 'subsequent'.

Line 117: I don't really understand this sentence. Are you simply trying to say that before the Jurassic (do you mean the Tr/J boundary?) there are broad similarities, and that afterwards the trajectories are largely divergent?

Line 118: I think the word 'contradictory' needs to replace with something else (very different?).

Line 119: I suspect there is a better word/phrase to replace 'resolutions'.

Line 124. Replace 'perturbated' with 'volatile'?

Line 130: Replace 'reinforced by' with 'found in'?

Line 132: Replace 'Besides' with 'Interestingly', or just delete it without replacement?

Line 138: Here is this fuzzy use of the Jurassic, which just seems a bit arbitrary, and a little odd since itself in an interval, not a point in time.

Line 138. Replace "Owing to" with 'Given the'.

Line 154: I don't see what the relationship is between the brachiopod diversity drop at the G-L boundary and their drop at the PTME – yes, both are big drops, but the PTME drop may well have been just as big even if there was no G-L drop.

Line 175. Period missing.

Lines 187-202. This analysis is particularly nice.

Lines 203-229: This analysis is also particularly nice – it lovely to see the authors use their data to tease apart (and refute) past claims – now I feel I am really learning something.

Line 221: Insert “They argue that” At the beginning of the new sentence on this line.

Line 233: I don't understand what you mean by 'predictable'. Do you mean simply that they make sense? Or do you just want to back off a little and begin this section by simply stating that 'For brachiopods there is a heterogeneity of rates between the different lifestyles.'?”

Line 246: Delete 'part'.

Line 247: Begin with 'of the'.

Lines 267-292: I am learning something again here, nice.

Line 294: Insert 'geographic' in here, so it is crystal clear that this about geography?

Line 312: What does MBD stand for?

Line 313: What four smaller time windows? I don't recall reading about these yet.

Line 317: Replace 'cannot' with 'may not'?

Line 318: Then maybe delete 'sufficiently'?

Lines 330-354. This is a little awkward (but it is OK). FYI, I came up with the same conclusions looking at your data (Fig.5) – nice analysis, and nice to see you have not fallen into the trap of a simplistic explanation.

Lines 355-362. Nice reasoning: this is what I wanted to know now.

Line 363. Delete 'Being'?

Lines 375-391. Well done; I find Liow's arguments 'awkward', and you have made explicit the 'awkwardness' very well, even though it is not quite resolved.

Line 424: I don't know what you mean by 'rare'.

Line 424: The word 'caused' feels like the wrong word. Correlated?

Line 424: Here you mention the long and short time window analysis, but I don't feel I have been told what this means. Oh, do you simply mean Phanerozoic (minus the Cambrian) versus Permian-to-Jurassic? Please make explicit.

Line 433: Do you mean '(or maybe even stimulated)'?

Line 444-445. Interesting.

Line 457: Replace 'could' with 'may'?

Line 466: Replace 'was not revealed' with 'not seen' (whatever short-time window analyses means)?

Line 510: Replace 'contradictory' with 'different'?

Line 511: It seems to me that a more accurate statement would be that before the Jurassic they show broadly similar diversification dynamics.

Line 520: And due to the loss of the PTe brachiopod group?

Reviewer #2 (Remarks to the Author):

The authors have satisfactorily addressed all of the comments from my previous review. I have no further comments or suggestions.

RESPONSE TO REVIEWERS' COMMENTS

Reviewer #1 (Remarks to the Author):

Summary: This is the second time I have seen this paper. I felt the first version was too diffuse, did not make proper contact with the literature, and coupled with my feeling that this issue was already settled, to warrant publication in this venue. However, upon reading this revision, I am convinced that this is indeed should be published in Nature Communications – it is a very good paper, with several sophisticated yet clearly expressed arguments relevant to establishing causality of diversity change in the fossil record, not just for bivalves versus brachiopods, but in general. This is an important paper, well executed.

Below are a series of largely minor comments that I think will help make the paper more readable (frankly, I don't know how anyone can write a sophisticated paper in another language, but these authors have done a fine job). In the interests of full disclosure, I did not scrutinize the methods section.

REPLY. Many thanks for your valuable comments and corrections which make our manuscript more readable.

Suggestions for improvement

Lines 21-22: The authors' say that "we find that unexpectedly the two clades displayed highly comparable trends of diversification before the Jurassic". It seems to me that the details are actually quite different (I don't see the 'highly comparable trends'), but they DO share similar same large-scale trends, so perhaps this generality ('similar large-scale trends') is the phrase that belongs in the abstract (not 'highly comparable trends').

REPLY. Thanks for your suggestion. We have revised it.

Line 38: The reference to the 'Big Five' (Raup and Sepkoski 1982) is over 40 years old; maybe the authors' could cite the 2023 review of the Big Five in the new journal "Cambridge Prisms: Extinction".

REPLY. Thanks. We have added this reference.

Line 79: 'dependence' mis-spelled.

REPLY. Corrected.

Line 92: Replace 'any difference arising between' with 'differences between'.

REPLY. Corrected.

Line 102: I still can't see the 'comparable trends of diversification dynamics, especially before the Jurassic' in the Figure (my eye is going to the fine detail of the wiggles). But if the next sentence explains what you mean, then I agree. So maybe just insert 'viz.'

between the sentences to indicate that you are going to tell is in what way they are similar – then I am OK with the statement.

REPLY. Thank you for your suggestions. We have modified it.

Line 116: Insert ‘their’ before ‘subsequent’.

REPLY. Corrected.

Line 117: I don't really understand this sentence. Are you simply trying to say that before the Jurassic (do you mean the Tr/J boundary?) there are broad similarities, and that afterwards the trajectories are largely divergent?

REPLY. Yes, we are simply saying that the large-scale trends are similar from the Ordovician to Triassic (before the Tr/J boundary), but the trajectories are largely divergent from the Jurassic to recent (after the Tr/J boundary). We have added some sentences to make it clear.

Line 118: I think the word ‘contradictory’ needs to replace with something else (very different?).

REPLY. Thanks. Corrected.

Line 119: I suspect there is a better word/phrase to replace ‘resolutions’.

REPLY. We have deleted this word. The word ‘volatility’ has already expressed our meaning.

Line 124. Replace ‘perturbated’ with ‘volatile’?

REPLY. Corrected.

Line 130: Replace ‘reinforced by’ with ‘found in’?

REPLY Corrected.

Line 132: Replace ‘Besides’ with ‘Interestingly’, or just delete it without replacement?

REPLY. Corrected.

Line 138: Here is this fuzzy use of the Jurassic, which just seems a bit arbitrary, and a little odd since itself in an interval, not a point in time.

REPLY. Thanks for your suggestion. We have addressed the Tr/J boundary in the revised text and re-worded this sentence.

Line 138. Replace ‘Owing to’ with ‘Given the’.

REPLY. Corrected.

Line 154: I don't see what the relationship is between the brachiopod diversity drop at the G-L boundary and their drop at the PTME – yes, both are big drops, but the PTME drop may well have been just as big even if there was no G-L drop.

REPLY. Sorry for the misleading. The reason why brachiopod diversity declined more

sharply is because the brachiopod extinction rate is much higher than that of bivalve in the PTME. We have deleted the statement about the G-L extinction.

Line 175. Period missing.

REPLY. Corrected.

Lines 187-202. This analysis is particularly nice.

REPLY. Thank you!

Lines 203-229: This analysis is also particularly nice – it lovely to see the authors use their data to tease apart (and refute) past claims – now I feel I am really learning something.

REPLY. Thank you for your encouragement.

Line 221: Insert “They argue that” At the beginning of the new sentence on this line.

REPLY. Revised.

Line 233: I don't understand what you mean by 'predictable'. Do you mean simply that they make sense? Or do you just want to back off a little and begin this section by simply stating that 'For brachiopods there is a heterogeneity of rates between the different lifestyles.'?

REPLY. The previous statement means that the heterogeneity can be easily predicted from their diversity changes. For example, the cemented and reclining brachiopods should have a higher extinction rate than the pedicle-attached ones in the PTME because many orders of them became extinct in the PTME. To make the statement clear and readable, we have deleted the word 'predictable' and simply stating the presence of rate heterogeneity.

Line 246: Delete 'part'.

REPLY. Corrected.

Line 247: Begin with 'of the'.

REPLY. Corrected.

Lines 267-292: I am learning something again here, nice.

REPLY. Thank you.

Line 294: Insert 'geographic' in here, so it is crystal clear that this about geography?

REPLY. Revised.

Line 312: What does MBD stand for?

REPLY. We mentioned this word in the previous 'regional analysis' section and therefore the abbreviation was used. Since this section is all about the result of MBD analysis, we have re-written the complete words (multivariate birth-death) at the

beginning of this section to make it clear.

Line 313: What four smaller time windows? I don't recall reading about these yet.

REPLY. The long-time window in our study refers to the entire Permian–Jurassic period. We then subdivided this long period into four shorter time windows (Asselian–Wordian, Capitanian–Ladinian, Carnian–Toarcian, Aalenian–Tithonian) to investigate if the correlations between rates and factors are changeable through time. Both long- and short-time window analyses are useful, as explained in the manuscript. The four short time windows were described in the Method section. We have added their names in the main text to make the discussion more readable.

Line 317: Replace 'cannot' with 'may not'?

REPLY. Corrected.

Line 318: Then maybe delete 'sufficiently'?

REPLY. Corrected.

Lines 330-354. This is a little awkward (but it is OK). FYI, I came up with the same conclusions looking at your data (Fig.5) – nice analysis, and nice to see you have not fallen into the trap of a simplistic explanation.

REPLY. Yes, it is bit difficult to explain the result, and we have tried to make our explanations clear and more understandable.

Lines 355-362. Nice reasoning: this is what I wanted to know now.

REPLY. Thanks.

Line 363. Delete 'Being'?

REPLY. Corrected.

Lines 375-391. Well done; I find Liow's arguments 'awkward', and you have made explicit the 'awkwardness' very well, even though it is not quite resolved.

REPLY. Thanks. The Reitan and Liow's multivariate analysis is also very complex, and multiple relationships were analysed at the same time. More studies are needed to explain their results.

Line 424: I don't know what you mean by 'rare'.

REPLY. 'Rare' means 'almost no'. Most of the factors don't show the selectivity of bivalves against brachiopods, but temperature is an exceptional factor.

Line 424: The word 'caused' feels like the wrong word. Correlated?

REPLY. Corrected.

Line 424: Here you mention the long and short time window analysis, but I don't feel I have been told what this means. Oh, do you simple mean Phanerozoic (minus the

Cambrian) versus Permian-to-Jurassic? Please make explicit.

REPLY. Following your suggestion, we have mentioned them at the beginning of the 'The role of competition' section.

Line 433: Do you mean '(or maybe even stimulated)'?

REPLY. Yes. Revised.

Line 444-445. Interesting.

REPLY: Thanks.

Line 457: Replace 'could' with 'may'?

REPLY. Revised.

Line 466: Replace 'was not revealed' with 'not seen' (whatever short-time window analyses means)?

REPLY. Revised. The short-time window has been explained.

Line 510: Replace 'contradictory' with 'different'?

REPLY. Revised.

Line 511: It seems to me that a more accurate statement would be that before the Jurassic they show broadly similar diversification dynamics.

REPLY. Thanks. Corrected.

Line 520: And due to the loss of the PTe brachiopod group?

REPLY. Revised.

Reviewer #2 (Remarks to the Author):

The authors have satisfactorily addressed all of the comments from my previous review. I have no further comments or suggestions.

REPLY. Thank you very much for reviewing our manuscript again and positive comments.